

# Update and evaluation of the ozone dry deposition in the Oslo CTM3 v1.0

Stefanie Falk[1] and Amund Søvde Haslerud[2,a]

[1]Department of Geosciences, University of Oslo, Oslo, Norway
[2]CICERO Center for International Climate Research, Oslo, Norway
[a]Kjeller Vindteknikk, Kjeller, Norway

**Correspondence:** Stefanie Falk (stefanie.falk@geo.uio.no)

**Abstract.** High concentrations of ozone in ambient air are hazardous not only to humans but to the ecosystem in general. The impact of ozone damage on vegetation and agricultural plants in combination with advancing climate change may affect food security in the future. While the future scenarios in themselves are uncertain, there are limiting factors constraining the accuracy of surface ozone modeling also at present: The distribution and amount of ozone precursors and ozone depleting substances, the

stratosphere-troposphere exchange as well as scavenging processes. Removal of any substance through gravitational settling or by uptake by plants and soil is referred to as dry deposition. The process of dry deposition is important for predicting surface ozone concentrations and understanding the observed amount and increase of tropospheric background ozone. The conceptual dry deposition velocities are calculated following a resistance-analogous approach wherein aerodynamic, quasi laminar, and canopy resistances are key components, but these are hard to measure explicitly. In this paper, we present an update of the dry

deposition scheme implemented in the Oslo CTM3. We change from a purely empirical dry deposition parameterization to a more process-based one which is taking the state of the atmosphere and vegetation into account. Examining the sensitivity of the scheme to various parameters, our focus lies mainly on the stomatal conductance-based description of the canopy resistance. We evaluate the resulting modeled ozone dry deposition with respect to observations and multi-model studies and also estimate the impact on the modeled ozone concentrations at the surface. We show that the global annual total ozone dry

deposition decreases with respect to the previous model version ($-47\%$), leading to an increase in surface ozone of up to $100\%$. While high sensitivity to changes in dry deposition to vegetation is found in the tropics, the largest impact on global scales is associated to changes in dry deposition to the ocean and deserts.

## 1 Introduction

Ozone is an important trace gas for all lifeforms on Earth. Depending on the place of its occurrence it has either a positive or

negative connotation. In the stratosphere, ozone absorbs most of the ultraviolet (UV)-light from the sun within the range of $100$–$315\,\mathrm{nm}$, thus shielding the Earth's surface from the most harmful UV-radiation. In addition, ozone is a potent greenhouse gas in both, stratosphere and troposphere. With a radiative forcing of $0.40 \pm 0.20\,\mathrm{W\,m^{-2}}$, it is placed third, only surpassed by $CO_2$ and $CH_4$ (IPCC - Intergovernmental Panel on Climate Change, 2013, Chapter 8).

In the troposphere and in particular in ambient air, ozone is considered as a highly toxic pollutant. Since the industrial revo-





lution, tropospheric background ozone concentrations have been increasing in the northern hemisphere (IPCC - Intergovernmental Panel on Climate Change, 2013, Chapter 2). In recent years, the number of episodes of peak concentrations has been decreasing in North America and Europe due to the implementation of air quality regulations (e.g., Wilson et al., 2012). At the same time, fast developing countries, like e.g., China or India, saw a significant increase in ozone related air pollution.

Continuously high concentrations of ambient air ozone are hazardous to the whole ecosystem. It is estimated that ozone is cause to an increase in pre-mature deaths (WHO - World Health Organization, 2008), an average global loss of yield in the four major crops (wheat, rice, maize, and soybean) of about $3 - 15\%$ (Ainsworth, 2017) as well as $7\%$ loss in primary production in forestry (Wittig et al., 2009; Matyssek et al., 2012). The impact of ozone damage on vegetation and agricultural plants may affect food security in the future especially in Asia (Tang et al., 2013; Tai et al., 2014; Chuwah et al., 2015) and might be an

important additional feedback to climate change (Sitch et al., 2007).

Tropospheric ozone is mainly produced in situ in complex photochemical cycles involving precursor gases such as carbon monoxide (CO) or volatile organic substances (VOCs – also known as hydrocarbons) in the presents of nitrogen oxides ($NO_x$). A typical reaction mechanism for CO is sketched in the following. In a sequence of rapid reactions a peroxyl radical $HO_2^\bullet$ is formed through an initial reaction of CO with a hydroxyl radical $^\bullet OH$. Via a reaction between $HO_2^\bullet$ and NO, $NO_2$

is formed which is then photolyzed. The resulting atomic oxygen reacts then with $O_2$ (and also under the presence of available co-reactants) to form an ozone molecule. Such a cycle leads to a net production via:

$$CO + 2O_2 + h\nu \rightarrow CO_2 + O_3. \tag{R1}$$

Similar cycles involving VOCs exist (Monks et al., 2015). Another source of tropospheric ozone is downward transport from the stratosphere via stratosphere-troposphere exchange (STE) (WMO - Global Ozone Research and Monitoring Project, 2014).

Based on observations, STE might only amount to roughly $10\%$ ($550 \pm 140\,\mathrm{Tg\,a^{-1}}$) of the total global ozone budget in the troposphere, while ozone from chemical production is estimated to be $5000\,\mathrm{Tg\,a^{-1}}$ (Monks et al., 2015). Ozone is removed from the atmosphere by photochemical reactions or scavenging processes. Major sinks are photolysis followed by a reaction with water vapor to from OH, reactions with $HO_2$, titration reactions, and dry deposition. We will come back to the latter later in this section and cover the implemented scheme in more detail in Section 2.1.

A dry deposition related sink limited to Arctic regions are so called ozone depleting events. They occur in spring-time in the polar boundary layer where an outburst of bromine monoxide BrO (so called bromine explosion) leads to a rapid depletion of surface ozone (Oltmans, 1981; Bottenheim et al., 1986; Barrie et al., 1988; Bottenheim and Chan, 2006). Various schemes ranging from bulk-snow parameterization (Toyota et al., 2011; Falk and Sinnhuber, 2018) to detailed in-snow (Toyota et al., 2014), and aerosol chemistry (Yang et al., 2010) have been successfully applied to different types of atmospheric models but

do not yet cover the full range of observed events. Although these ozone depleting events are important to understand surface ozone abundance in Arctic regions, we have not implemented any parameterization of these processes in the Oslo CTM3 as of now. We also do not consider the contribution of very short-lived ozone depleting substances (VSLS), that affect stratospheric ozone (Warwick et al., 2006; Ziska et al., 2013; Hossaini et al., 2016; Falk et al., 2017), to the tropospheric ozone abundance. Since ozone is highly reactive, its global mean life-time in the troposphere is roughly 22 days but ranges between a few





days in the tropical boundary layer to up to 1 year in the upper troposphere (Stevenson et al., 2005; Young et al., 2013). The abundance of tropospheric ozone therefore varies, e.g., with time of the day (maximum ~15:00 local time), season (mid-June maximum), altitude, location (Schnell et al., 2015), or weather conditions in general (Otero et al., 2018). Typical concentrations of surface ozone range from $10\,\mathrm{ppb}$ over the tropical Pacific to $100\,\mathrm{ppb}$ in the downwind areas of highly emitting sources

(IPCC - Intergovernmental Panel on Climate Change, 2013, Chapter 8). This variability poses a challenge on both, trend analysis from observation as well as validation and intercomparison of models. At the observational side, there is only a limited number of long-term ozone observations, mainly restricted to European sites. Most of these indicate a doubling of tropospheric ozone since the 1950s (IPCC - Intergovernmental Panel on Climate Change, 2013, Chapter 2). But especially the very low pre-industrial ozone abundance cannot be reproduced by the likes of most models. Among the participating models in the

Atmospheric Chemistry and Climate Model Intercomparison Project (ACCMIP), there is a general tendency to underestimate tropospheric ozone burden (e.g., $10-20\,\%$ negative bias at $250\,\mathrm{hPa}$ in the southern hemisphere (SH) tropical region) (IPCC - Intergovernmental Panel on Climate Change, 2013, Chapter 8). With respect to surface ozone, Schnell et al. (2015) conclude that all ACCMIP models, which reported hourly surface ozone, tend to overestimate surface ozone values in North America and Europe in comparison with available observations. A key to fathom these slightly contradicting results may lie in the used

dry deposition schemes.

Removal of any substance through gravitational settling or by uptake by plants and soil is referred to as dry deposition. The process of dry deposition is important for predicting surface ozone concentrations and understanding the observed amount and increase of tropospheric background ozone. It is estimated that about $1000 \pm 200\,\mathrm{Tg\,a^{-1}}$ of ozone are removed from the atmosphere by dry deposition processes (Monks et al., 2015). Conceptually, dry deposition is a product between surface

ozone concentration $[\mathrm{O_3}](z_0)$ and a dry deposition velocity $v_{\mathrm{DD}}^{\mathrm{O_3}}$. Species dependent dry deposition velocities $v_{\mathrm{DD}}^{i}$, which are synonymously referred to as conductance $G^i$, for any gaseous species $i$, are typically calculated following a resistance-analogous approach

$$v_{\mathrm{DD}}^{i} = \frac{1}{R_a + R_b^i + R_c^i},\tag{1}$$

wherein aerodynamic $R_a$, quasi-laminar layer $R_b^i$, and canopy resistances $R_c^i$ are key components (Wesely, 1989; Seinfeld and

Pandis, 2006). For all gases, $R_a$ is the same, while $R_b^i$ and $R_c^i$ vary from gas to gas and also depend on land surface types (e.g., ice/snow, water, urban, desert, agricultural land, deciduous forest, coniferous forest etc.). Originally, Wesely (1989) used fixed seasonal average dry deposition resistances for each land surface type. For all three types of resistances in this Wesely-type parameterization, more process-oriented formulations have been developed and validated over the years. Luhar et al. (2017) have validated ozone dry deposition to the ocean with respect to three different formulations of surface resistances. An update

on the ozone surface resistance over snow and ice covered surfaces has been provided from combined model and observation studies (Helmig et al., 2007, $v_{\mathrm{ice/snow}}^{\mathrm{O_3}} = 1/10000\,\mathrm{m\,s^{-1}}$). Canopy conductance is parameterized at the single-leaf-level (stomatal conductance) for various plant function types (PFT) as well as for single plant species based on empirical studies (Jarvis, 1976; Ball et al., 1987; Simpson et al., 2012; Mills et al., 2017). But progress has also been made on process-oriented modeling of stomatal conductance (Anderson et al., 2000; Buckley, 2017). The variety of differing formulations and choices of parame-



ters leads to a wide spread of results in model intercomparisons (Hardacre et al., 2015; Derwent et al., 2018) and about $20\%$ uncertainty on the resulting total dry deposition (Monks et al., 2015).

In Section 2, we will briefly describe the Oslo CTM3, give a detailed account of the new dry deposition scheme (Section 2.1) as well as present pre-processing of meteorological input data to compute necessary input to the dry deposition scheme such as begin and duration of greening season (GDAY, GLEN) and photosynthetic photon flux density (PPFD) (Section 2.3). In Section 3, we present sensitivity tests with respect to a manifold of parameters in the dry deposition scheme (Section 3.1) and validate our results with respect to results from the multi-model intercomparison of Hardacre et al. (2015) (Section 3.2) and to observations at the surface (Section 3.3). In Section 4, we will summarize and discuss our results and draw conclusions for further development of the model.

## 2   Model description

The Oslo CTM3 is a three dimensional global chemistry transport model (CTM). The key components of the Oslo CTM3 have been described and evaluated by Søvde et al. (2012). A detailed account of the capabilities of the Oslo CTM3 in simulating anthropogenic aerosol forcing in the past and recent past using the Community Emission Data System (CEDS) historical emission inventory (Hoesly et al., 2018) is given by Lund et al. (2018). The Oslo CTM3 can also be coupled to the Model of Emissions of Gases and Aerosols from Nature (MEGAN v2.10) (Guenther et al., 2006). A publication focusing on this is planed. While the meteorological data driving the Oslo CTM3 is given in a resolution of T159N80L60, with the highest model level at $0.02\,\mathrm{hPa}$, it is very time and memory consuming to run the Oslo CTM3 with full chemistry at this resolution. Therefore, we reduced the horizontal resolution to $2.25° \times 2.25°$ in our experiments. In the following, we will give a detailed account of the new dry deposition scheme. Although some of the equations in this section may be *textbook knowledge*, for the sake of completeness, it is important to summarize them, nonetheless.

### 2.1   Ozone dry deposition scheme

In the original dry deposition scheme of the Oslo CTM3, the state of the atmosphere was not taken into account. Dry deposition velocities were rather parameterized following the work of Wesely (1989) with parameter updates from Hough (1991). This means that seasonal day and night average deposition velocities for different land surface types (water, forest, grass, tundra/desert, and ice and snow) were in use. Day was distinguished from night by solar zenith angles below $90°$. Winter was defined by temperatures below $273.15\,\mathrm{K}$ for gridboxes containing land masses. For ocean, winter and summer parameters are equal in this parameterization, therefore no distinctive treatment is needed for ocean gridboxes. In addition, a reduced uptake due to snow cover above $1\,\mathrm{m}$ for forest and $10\,\mathrm{cm}$ for grass/tundra, respectively, was taken into account. We will refer to this parameterization as *Wesely scheme*.




Regarding the new dry deposition scheme, we follow the European Monitoring and Evaluation Programme (EMEP) MSC-W model (Simpson et al., 2003, 2012), which is used for air quality modeling implementing the Convention on Long-Range Transboundary Air Pollution (CLRTAP). We will refer to this as *EMEP scheme* throughout the rest of the paper. The EMEP scheme is a more physical approach compared to the previously used Wesely scheme, because it takes state (e.g., pressure, tem-

perature) of the atmosphere as well as dynamics (e.g., wind stress) of the boundary layer into account. To a certain degree, the global variety of plants and their variability throughout the seasons is also acknowledged. The EMEP scheme is implemented for the gaseous species $O_3$, $H_2O_2$, $NO_2$, PAN, $SO_2$, $NH_3$, HCHO, and $CH_3CHO$. Since CO has a very small uptake and is not included in Simpson et al. (2003, 2012), the Wesely parameterization is kept. In addition to the gaseous species, some of the aerosol deposition velocities are also modified, namely black carbon (BC) and organic carbon (OC), sulfuric aerosols

($SO_4$, MSA) and secondary organic aerosols (SOA).

As displayed in Eq. (1), the dry deposition computation is typically split into three different resistance contributions (aerodynamic $R_a$, quasi-laminar layer $R_b^i$, and canopy $R_c^i$) which we will recap in the following.

### 2.1.1 Aerodynamic resistance

The aerodynamic resistance is describing the turbulent transport of any substance down to the surface. In Simpson et al. (2003,

2012) it is described as

$$R_a = \frac{1}{\kappa u_*} \left[ \ln\left(\frac{z-d}{z_0}\right) - \Psi_m\left(\frac{z-d}{z_0}\right) - \Psi_m\left(\frac{z_0}{L}\right) \right], \quad (2)$$

with the Kármán constant $\kappa = 0.40$, the friction velocity $u_*$, integrated stability equations for momentum $\Psi_m$, a constant $d$ (typically $0.7\,\mathrm{m}$), and the Obukhov length $L$. For certain values of $z$, $z_0$, and $L$, this may result in nonphysical (negative) values for $R_a$. For this reason, we diverge from the EMEP scheme at this point and fall back to the method of Monteith (1973),

wherein the sensible heat flux $\Phi_{Q_{\text{sensible}}}$ is written as

$$\Phi_{Q_{\text{sensible}}} = \rho_{\text{air}} c_P \cdot \frac{\partial_z T}{\partial_z R_a}. \quad (3)$$

Herein, $\rho_{\text{air}}$ is the air density, $c_P$ the specific heat at constant pressure $P$, $\partial_z R_a$ is the diffusion resistance to sensible heat between surface ($z_0 = 0$) and the reference height $z$, and $\partial_z T$ is the difference between the surface temperature $T_0$ and the temperature at a reference height $T_z$. From the eddy diffusion theorem, a similar formulation can be derived

$$\Phi_{Q_{\text{sensible}}} = \rho_{\text{air}} c_P \cdot \frac{\partial_z T}{\partial_z u} \cdot u_*^2. \quad (4)$$

Comparing Eq. (3) and Eq. (4) and assuming $\partial_z R_a \to R_a$ and $\partial_z u \to u_z$ for a finite $z$, we find

$$R_a = \frac{u_z}{u_*^2}. \quad (5)$$

As reference height $z$, we use the height at mid-level of the lowermost model level (roughly $8\,\mathrm{m}$). From the meteorological input fields $u_z$ is directly available, while $u_*$ can be computed via

$$u_* = \sqrt{\frac{\tau_0}{\rho}}, \quad (6)$$

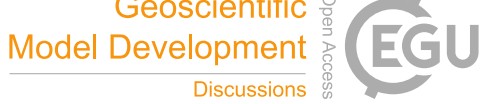



with the turbulent surface stress $\tau_0^2 = \mathrm{NSSS}^2 + \mathrm{EWSS}^2$, and northward turbulent surface stress (NSSS), eastward turbulent surface stress (EWSS).

### 2.1.2 Quasi-laminar layer resistance

The quasi-laminar layer resistance $R_b^i$ is species specific and differs over land and ocean surfaces. In case a gridbox contains both, a weighted mean is calculated. Over land, we use (Eq. (53), Simpson et al., 2012)

$$R_b^i = \frac{2}{\kappa u_*} \cdot \left(\frac{\mathrm{Sc}_i}{\mathrm{Pr}}\right)^{\frac{2}{3}}, \tag{7}$$

wherein Pr is the Prandtl number (typically 0.72 for air and other gases) and $\mathrm{Sc}_i$ is the Schmidt number for a gas $i$. Eq. (7) differs from a similar formulation in Seinfeld and Pandis (2006) by a factor of roughly 1.25. From $\mathrm{Sc}_i = \nu/D_i$, with the kinematic viscosity of air $\nu$, we derive a Schmidt number in water equivalent:

$$\mathrm{Sc}_i = \frac{D_{\mathrm{H_2O}}}{D_i} \cdot \mathrm{Sc}_{\mathrm{H_2O}}, \tag{8}$$

with the molecular diffusivity for any gas $D_i$, the Schmidt number of water ($\mathrm{Sc}_{\mathrm{H_2O}} = 0.6$) and its molecular diffusivity ($D_{\mathrm{H_2O}} = 0.21 \cdot 10^{-4}\,\mathrm{m^2\,s^{-1}}$). The used ratios $D_{\mathrm{H_2O}}/D_i$ are tabulated in Supplement S.1. Over ocean, we use (Eq. (54), Simpson et al., 2012)

$$R_b^i = \frac{1}{\kappa u_*} \cdot \ln\left(\frac{z_0}{D_i} \cdot \kappa u_*\right) \tag{9}$$

with an imposed lower threshold of $10\,\mathrm{s\,m^{-1}}$ and an upper limit of $1000\,\mathrm{s\,m^{-1}}$. The computation of roughness length $z_0$ over ocean is divided into a *calm* and a *rough* sea case, with a threshold of $3\,\mathrm{m\,s^{-1}}$. For calm sea, we apply the following upper limit (Hinze, 1975; Garratt, 1992, with a slightly higher coefficient of 0.135)

$$z_0^{\mathrm{calm}} = \min\left\{2 \cdot 10^{-3}, 0.135 \cdot \frac{\nu}{u_*}\right\}. \tag{10}$$

The kinematic viscosity of air $\nu$ herein can be computed from

$$\nu = \frac{\mu}{\rho} = \frac{\mu(T)}{\frac{P_0}{T \cdot R_{\mathrm{air}}}}. \tag{11}$$

For the temperature dependent dynamic viscosity of air $\mu(T)$, we chose a linear fit to Sutherland's law through the origin within the temperature range $\{T \in \mathbb{R} | (243.15 < T < 313.15)\,\mathrm{K}\}$: $\mu(T) = 6.2 \cdot 10^{-8}\,\mathrm{kg\,m^{-1}\,s^{-1}\,K^{-1}} \cdot T$. But despite its rough accuracy, we found that the choice of $\mu(T)$ has no effect on $\overline{R}_b$ (Supplement S.2: Figs. S1–S2). In Eq. (11), $\rho$ is substituted by the air density using the ideal gas law. $P_0$ is the surface pressure, as $T$ the $2\,\mathrm{m}$ temperature is chosen, and $R_{\mathrm{air}}$ is the universal gas constant for air. The rough sea case follows the method of Charnock (1955); Wu (1980):

$$z_0^{\mathrm{rough}} = \min\left\{2 \cdot 10^{-3}, 0.018 \cdot \frac{u_*^2}{g}\right\} \tag{12}$$

with a gravitational acceleration $g = 9.836\,\mathrm{m\,s^{-2}}$. The allowed maximum roughness length in both cases is set to $2\,\mathrm{mm}$. Since the $z_0$ computed with this parameterization are rather small ($0 < z_0^{\mathrm{calm}} < 1 \cdot 10^{-4}\,\mathrm{m}$, $0 < z_0^{\mathrm{rough}} < 2 \cdot 10^{-3}\,\mathrm{m}$), $R_b^i$ is set to its lower limit of $10\,\mathrm{s\,m^{-1}}$ in about $91\,\%$ of all cases (see Supplement S.2: Fig. S3).



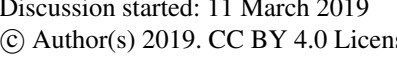
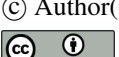

### 2.1.3 Surface resistance

The surface resistance consists of both, stomatal and non-stomatal resistances. When computing gridbox averages, we use the sum of conductances (deposition velocities)

$$G_c = \text{LAI} \cdot G_{\text{sto}} + G_{\text{ns}}, \tag{13}$$

wherein LAI is the leaf area index (zero for non-vegetated surfaces), $G_{\text{sto}}$ the stomatal conductance, and $G_{\text{ns}}$ the non-stomatal conductance.

The stomatal conductance is a measure of the rate of $CO_2$ exchange and evapotranspiration through the stomata of a leaf. There are several environmental conditions affecting the opening and closing of the stomata and hence their capability of respiration (e.g., light, available water, etc.). Stomata sluggishness, a state in which the stomata can no longer fully close, has

been reported as ozone induced damage (Hoshika et al., 2015), but is not taken into account in our formulation. To reflect part of the underlying mechanism, the stomatal conductance $g_{\text{STO}}$ in the EMAP scheme is computed using a multiplicative ansatz that is also explained in much detail in Mills et al. (2017):

$$g_{\text{STO}} = g_{\text{max}}^{\text{m}} \cdot f_{\text{phen}} \cdot f_{\text{light}} \cdot \max\{f_{\text{min}}, f_T \cdot f_D \cdot f_{\text{SW}}\}. \tag{14}$$

The factors herein are normalized and vary within the range $0-1$. They account for leaf phenology ($f_{\text{phen}}$), light ($f_{\text{light}}$),

temperature ($f_T$), water vapor pressure deficit ($f_D$), and soil water content ($f_{\text{SW}}$).

The temperature adjustment $f_T$ of plants is computed from

$$f_T = \frac{T_{\text{2m}} - T_{\text{min}}}{T_{\text{opt}} - T_{\text{min}}} \cdot \left( \frac{T_{\text{max}} - T_{\text{2m}}}{T_{\text{max}} - T_{\text{opt}}} \right)^{\beta}, \tag{15}$$

with $\beta = \frac{T_{\text{max}} - T_{\text{opt}}}{T_{\text{opt}} - T_{\text{min}}}$. The parameters $T_{\text{min}}$, $T_{\text{max}}$ and $T_{\text{opt}}$ are tabulated for various land surface types. All parameters used in the EMEP scheme can be found in the supplementary to Simpson et al. (2012) or tabulated in Supplement S.3. Since $f_T$ turns

negative outside its range defined by $T_{\text{min}}$, $T_{\text{max}}$, we impose a lower limit of $0.01$.

The water vapor deficit (VPD) is proportional to the saturation partial pressure of water ($P_{\text{H}_2\text{O}}^s$) and relative humidity (RH)

$$\text{VPD} = P_{\text{H}_2\text{O}}^s \cdot (1 - \text{RH}/100). \tag{16}$$

Using tabulated values of $f_{\text{min}}$, $D_{\text{min}}$, $D_{\text{max}}$, the water vapor pressure deficit penalty factor $f_D$ can be computed:

$$f_D = f_{\text{min}} + (1 - f_{\text{min}}) \cdot \frac{D_{\text{min}} - \text{VPD}}{D_{\text{min}} - D_{\text{max}}}. \tag{17}$$

The penalty factor with respect to available soil water (SW) $f_{\text{SW}}$ is defined as

$$f_{\text{SW}} = \begin{cases} 1 & \text{if} \quad \text{SW} \geq 0.5, \\ 2 \cdot \text{SW} & \text{if} \quad \text{SW} < 0.5. \end{cases} \tag{18}$$





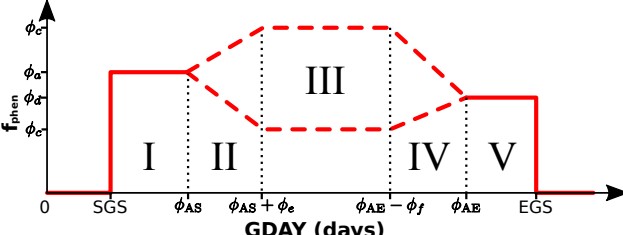

**Figure 1.** Sketch of the five different phases in plant phenology $f_{\text{phen}}$ in accordance to Eq. (19).

Simpson et al. (2012) suggest using SW of soil depths below $1\,\text{m}$ (SWVL4 in OpenIFS). Throughout most of our simulations (Section 3), we have, however, used the soil water contained in the top layer (SWVL1, $0 - 7\,\text{cm}$), which makes the canopy conductance more sensitive to precipitation than it would be otherwise.

The phenology of a plant typically describes its life-cycle throughout a year, e.g., at mid latitudes and for deciduous species, it starts with the emergence of leafs in spring and ends in fall. In the EMEP scheme, phenology is parameterized with respect to the start of the greening season (SGS) and its end (EGS). Details about our treatment of these are given in Section 2.3.1. In summary, our adaption of the $f_{\text{phen}}$ parameterization reads as follows:

$$
f_{\text{phen}} = \begin{cases}
if \quad \text{GDAY} = 0 & 0 \\
if \quad \text{GLEN} \geq 365 & 1 \quad \textit{(explicitly excluding tropics)} \\
else & \begin{cases}
if \quad \text{GDAY} \leq \phi_{\text{AS}} & \phi_a \\
if \quad \text{GDAY} \leq \phi_{\text{AS}} + \phi_e & \phi_b + (\phi_c - \phi_b) \cdot (\text{GDAY} - \phi_{\text{AS}})/\phi_e \\
if \quad \text{GDAY} \leq \text{GLEN} - \phi_{\text{AE}} - \phi_f & \phi_c \\
if \quad \text{GDAY} \leq \text{GLEN} - \phi_{\text{AE}} & \phi_d + (\phi_c - \phi_d) \cdot (\text{GLEN} - \phi_{\text{AE}} - \text{GDAY})/\phi_f \\
else & \phi_d
\end{cases}
\end{cases}
\tag{19}
$$

Herein, we use the SGS and EGS derived parameters day of greening season (GDAY), the time elapsed starting at the SGS, and the total length of the greening season (GLEN), the time span between EGS and SGS. The parameters $\phi_a$, $\phi_b$, $\phi_c$, and $\phi_d$ define start or end points in the five phases of phenology in the EMEP scheme, while $\phi_e$, $\phi_f$, $\phi_{\text{AS}}$, and $\phi_{\text{AE}}$ control the temporal timing (Fig. 1). If GLEN is zero we are, e.g., in Arctic regions and there is no vegetation anyway, therefore $f_{\text{phen}} = 0$. Before the start of the growing season (GDAY $= 0$) $f_{\text{phen}} = 0$. Since the EMEP scheme is tuned to northern hemisphere (NH) mid latitudes, this phenology does not apply to the tropics, after evaluation of the model results (Section 3.1), we decided to set $f_{\text{phen}} = 1$ if GLEN is greater or equal to 365 which is the case in the tropics.

Light in the wavelength band $400 - 700\,\text{nm}$ to which the plant chlorophyll is sensitive is called photosynthetic active radiation (PAR). The integral of PAR over these wavelengths is the photosynthetic photon flux density (PPFD). In the EMEP parameterization, this integrated photon flux is used to mimic the time lag of the opening and closing of stomata with respect

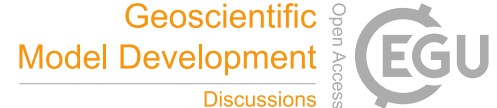



to light instead of a simple day-on/night-off treatment:

$$f_{\text{light}} = 1 - \exp(-\alpha_{\text{light}} \cdot \text{PPFD}). \tag{20}$$

Since $g_{\text{max}}$ in Eq. (14) is in units of $\text{mmol s}^{-1}\,\text{m}^{-2}$, a unit conversion is done:

$$g_{\text{sto}}^{\text{m}}(N) = g_{\text{sto}}(N) \cdot R \cdot \frac{T_0}{P_0}. \tag{21}$$

Herein, $R$ is the universal gas constant. With these, the total gridbox stomatal conductance $G_{\text{sto}}$ is computed weighted by the land surface type fraction $f_L(N)$:

$$G_{\text{sto}} = \sum_{N=0}^{N_{\text{max}}} f_L(N) \cdot g_{\text{sto}}^{\text{m}}(N) \tag{22}$$

In the EMEP scheme, non-stomatal conductances are explicitly calculated for $O_3$, $SO_2$, $HNO_3$, and $NH_3$. For all other species, an interpolation between $O_3$ and $SO_2$ values is carried out.

The non-stomatal conductance for $O_3$ consists of two terms, one depending on vegetation type and one depending on the soil/surface. For a land surface type $N$, we can write

$$G_{\text{ns}}^{O_3}(N) = \frac{\text{SAI}(N)}{r_{\text{ext}}} + \frac{1}{R_{\text{inc}}(N) + R_{\text{gs}}^{O_3}(N)}. \tag{23}$$

$\text{SAI}(N)$ is the surface area index for vegetation type $N$, which is LAI plus some value representing cuticles and others surfaces. The external leaf resistance is defined by

$$r_{\text{ext}} = 2000\,\text{s m}^{-1} \cdot F_T. \tag{24}$$

Herein $F_T$ is a temperature correction factor for temperatures below $-1\,^\circ\text{C}$ and $\{F_T \in \mathbb{R} | (1 \leq F_T \leq 2)\}$

$$F_T = \exp\left(-0.2 \cdot (1 + \theta_0)\right). \tag{25}$$

$\theta_0$ is the surface or $2\,\text{m}$ temperature in $^\circ\text{C}$. In general, $\text{SAI} \equiv \text{LAI}$ for all land types, except for:

$$\text{SAI} = \begin{cases} \text{LAI} + 1 & \textit{if forest / wetland}, \\ \text{LAI} \cdot 5/3.5 & \textit{if cropland}, \text{I. part of growing season}, \\ \text{LAI} + 1.5 & \textit{if cropland}, \text{II. part of growing season}, \\ 0 & \textit{if cropland}, \text{winter}. \end{cases} \tag{26}$$

Extending the EMEP scheme to the southern hemisphere, the used growing season for crops is defined in Table 1.

In this way, vegetation affects the conductance also by being there, not only by uptake through the stomata. The in-canopy resistance $R_{\text{inc}}$ is defined for each vegetated land type $N$ as

$$R_{\text{inc}} = b \cdot \text{SAI}(N) \cdot \frac{h(N, \text{lat})}{u_*}, \tag{27}$$





**Table 1.** Definition of growing season for crops used in the Oslo CTM3 in northern hemisphere (NH) and southern hemisphere (SH).

|      | I. part (days) | II. part (days) |
| ---- | -------------- | --------------- |
| NH   | 90–140         | 141–270         |
| SH   | 272–322        | 323–452         |

where $h(N, \text{lat})$ is the latitude dependent vegetation height (see explanation at the end of this section) and $b = 14\,\mathrm{s}^{-1}$ is an empirical constant. According to Simpson et al. (2012), the canopy resistance in EMEP does not take temperature and snow into account and is zero for non-vegetated surfaces.

$R_{\text{gs}}^{\text{O}_3}(N)$ is tabulated and corrected for temperature by $F_T$ and snow cover fraction $f_{\text{snow}}$:

$$\frac{1}{R_{\text{gs}}^{\text{O}_3}(N)} = \frac{1 - f_{\text{snow}}}{\hat{R}_{\text{gs}}^{\text{O}_3}(N)} + \frac{f_{\text{snow}}}{R_{\text{snow}}^{\text{O}_3}}. \tag{28}$$

As initially mentioned, the necessary depth of snow to cover a certain type of vegetation differs. Therefore, we calculate a snow cover fraction $f_{\text{snow}}$ using the snow depth $S_D$, which is available in units of meter of water equivalent from the meteorological input data, scaled to $10\,\%$ of the vegetation height.

The total conductance $G_{\text{ns}}^{\text{O}_3}$ in each gridbox is the sum of all $G_{\text{ns}}^{\text{O}_3}(N)$ weighted by the land surface type fraction $f_L(N)$

$$G_{\text{ns}}^{\text{O}_3} = \sum_{N=0}^{N_{\text{max}}} G_{\text{ns}}^{\text{O}_3}(N) \cdot f_L(N). \tag{29}$$

### 2.1.4 Latitude dependent vegetation height

The vegetation height $h(N, \text{lat})$ in the EMEP scheme is linearly decreasing with latitude between $60^\circ$ and $74^\circ$N. To adapt this to a global model, we need to make a few additional assumptions. The tabulated height for each vegetation type $h(N)$ in the EMEP scheme is regarded as constant at mid latitudes ($40^\circ - 60^\circ$). Towards the poles, we decrease the height of each vegetation type using the same rate as described in Simpson et al. (2012). At a latitude of $74^\circ$ a minimum height of $3/10 \cdot h(N)$ is reached and kept constant. Towards the equator, we increase the height linearly so that at a latitude of $10^\circ$ a maximum height of $2 \cdot h(N)$ is achieved which is then held constant. We also assume symmetry in both hemispheres. Presuming a typical tree height of $20\,\mathrm{m}$ at mid latitudes, this step-wise function yields a height of $8\,\mathrm{m}$ at high latitudes and $40\,\mathrm{m}$ in the tropics which is not unrealistic. For four example PFTs, results are shown in the Supplement (S.4, Fig. S4).

### 2.1.5 Mapping of land surface types

The Oslo CTM3 is configured to read land surface types from, either ISLSCP2 product from MODIS or Community Land Model (CLM) 2 categories, which have to be mapped to the nine land surface types used in the EMEP scheme (Fig. 2). For both, MODIS and CLM 2 land surface categories, snow and ice cover is estimated from input meteorology, while water is



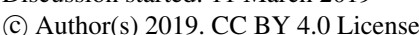


defined as $1 - \sum_{N=0}^{N_{\max}} f_L(N)$. From the MODIS category *Barren or sparsely vegetated*, everything polward from $60\,^\circ$ is defined as tundra, while everything equatorward is categorized as desert. This mapping differs from the one used in the Wesely scheme.

## 2.2 Dry deposition scheme for other gases and aerosols

For $NH_3$, $SO_2$, and $HNO_3$, we follow exactly the treatment as described by Simpson et al. (2012). Since the dry deposition velocity for CO is very small, we use a prescribed value for $v_{DD}^{CO}$ of $0.03\,\mathrm{cm\,s^{-1}}$.

The EMEP scheme for aerosols (BC, OC, $SO_4$, MSA, SOA) defines the surface deposition velocity $V_{ds}$ as

$$
\frac{V_{ds}}{u_*} = 
\begin{cases}
a_1 & if \quad L \geq 0\,\mathrm{m}, \\
a_1 \cdot F_N \left[ 1 + \left(\frac{a_2}{25}\right)^{2/3} \right] & if \quad -25\,\mathrm{m} < L < 0\,\mathrm{m}, \\
a_1 \cdot F_N \left[ 1 + \left(\frac{-a_2}{L}\right)^{2/3} \right] & if \quad L \leq -25\,\mathrm{m},
\end{cases}
\tag{30}
$$

wherein $F_N = 3$ for fine nitrate and ammonium and $F_N = 1$ for all other species. $L$ is again the Obukhov length. $a_2 = 300\,\mathrm{m}$, while $a_1$ differs for forest and non-forest. To account for hydrophilic and hydrophobic BC/OC aerosols on dry and wet surfaces, we diverge slightly from the EMAP scheme in the definition of the parameter $a_1$. From pre-studies with an aerosol microphysic model, we find

$$
a_1^{\text{hydrophob.}} = 0.025\,\mathrm{cm\,s^{-1}} \cdot \overline{u_*}
\tag{31}
$$

and

$$
a_1^{\text{hydrophil.}} = 0.2\,\mathrm{cm\,s^{-1}} \cdot \overline{u_*},
\tag{32}
$$

with the annual mean friction velocity $\overline{u_*}$. Our definition of $a_1$ is then:

$$
a_1 = 
\begin{cases}
if \quad land & 
\begin{cases}
if \quad non\text{-}forest \quad a_1^{\text{hydrophob.}} \\
if \quad forest \quad 
\begin{cases}
if \quad dry \quad \max\left\{ a_1^{\text{hydrophob.}}, 0.008 \cdot \frac{\mathrm{SAI}}{10} \right\} \\
if \quad wet \quad 1.5 \cdot \max\left\{ a_1^{\text{hydrophob.}}, 0.008 \cdot \frac{\mathrm{SAI}}{10} \right\}
\end{cases}
\end{cases} \\
if \quad ocean & 
\begin{cases}
a_1^{\text{hydrophil.}} \\
a_1^{\text{hydrophob.}}
\end{cases} \\
if \quad ice & 
\begin{cases}
if \quad \theta \leq 0\,^\circ\mathrm{C} \quad a_1^{\text{hydrophob.}} \\
if \quad \theta > 0\,^\circ\mathrm{C} \quad 
\begin{cases}
a_1^{\text{hydrophil.}} \\
a_1^{\text{hydrophob.}}
\end{cases}
\end{cases}
\end{cases}
\tag{33}
$$

The total aerosol surface deposition is defined as

$$
V_{ds}^{tot} = \sum_{N=0}^{N_{\max}} V_{ds}(N) \cdot f_L(N).
\tag{34}
$$

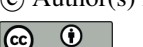



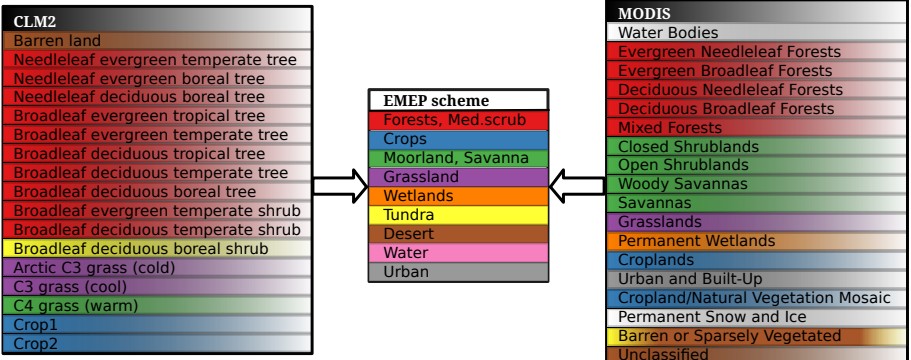

**Figure 2.** Mapping of land surface categories. Either land surface categories from ISLSCP2 product of MODIS / Community Land Model (CLM) 2 can be chosen for mapping to the nine land surface types used in the EMAP scheme. Water bodies and snow and ice categories of MODIS are actually not mapped. For both, MODIS and CLM 2 land surface categories, snow and ice cover is estimated from input meteorology, while water is defined as $1 - \sum_{N=0}^{N_{\max}} f_L(N)$. From MODIS category *Barren or sparsely vegetated*, everything polward from $60°$ is defined as tundra, while everything equatorward is categorized as desert.

## 2.3 Pre-processing

As mentioned in the previous section, there are two variables needed for computing the stomatal conductance which are not directly available from the meteorological input data. The greening season, as the time of the year in the mid and high latitudes when it is most likely for plants to grow, and the photosynthetic photon flux density, as the amount of light that plants need to photosynthesize. In the following, we present the necessary pre-processing of the variables. It is planed to implement an online computation of these variables into the Oslo CTM3 later on.

### 2.3.1 Greening season

In Eqs. (23–26), Simpson et al. (2012) use prescribed start of growing season (SGS) and end of growing season (EGS) at $50°$N ($d_{\text{SGS}}$, $d_{\text{EGS}}$) together with lapse rates ($\nabla d_{\text{SGS}}$, and $\nabla d_{\text{EGS}}$) to define phenology and dry deposition over agricultural areas. For the growing season of crops in the computation of non-stomatal conductances, we use also prescribed values (Table 1), while for the stomatal conductances, as shown in Eq. (19), we use the SGS and EGS derived parameters: day of greening season (GDAY), the time elapsed starting at the SGS, and the total length of the greening season (GLEN), the time span between EGS and SGS. Since the eurocentric EMEP parameterization of SGS and EGS is not applicable in a global model, another latitude dependent parameterization is needed. First, we have used a parameterization as implemented in the Sparse Matrix Operational Kernel Emissions – Biogenic Emission Inventory System (SMOKE-BEIS; model webpage). SMOKE-BEIS has fixed values for SGS and EGS for all regions but NH mid latitudes ($23° < \text{lat} < 65°$), where it uses lapse rates of $\nabla d_{\text{SGS}} = 4.5$ and $\nabla d_{\text{EGS}} = 3.3$. As this parameterization is optimized for North America, it does not work well in Europe, e.g., most of





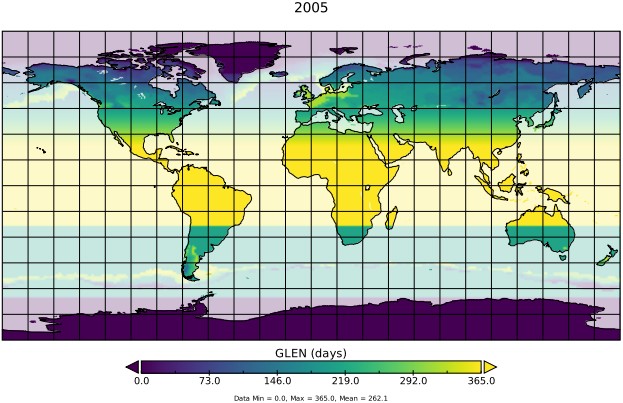

**Figure 3.** Pre-processing of greening season from meteorological surface temperature fields. Shown is the total length of the greening season (GLEN) for the year 2005. The $5\,^{\circ}$C-days criteria has been used in both hemispheres mid–high latitudes. Ocean has been shaded to indicate that greening season will only affect land.

northern Scandinavia has no allocated vegetation period.

In agriculture, there are different empirical rules to estimate the SGS and EGS. The simplest assumption is that greening starts after 5 consecutive days with a daily average temperature above $5\,^{\circ}$C and vice versa for EGS. Other estimates use growing

degree days (Levis and Bonan, 2004; Fu et al., 2014a), include soil moisture (Fu et al., 2014b), or rely on satellite observations. A comprehensive evaluation of different techniques is given by Anav et al. (2017).

Based on the empirical rule ($5\,^{\circ}$C-days), we have pre-processed our meteorological input data offline. We added some additional criteria to prevent for *false spring*: If, within these 5 days, the average temperature drops below or rises above $5\,^{\circ}$C, the counter is reset, respectively. First, we used the $5\,^{\circ}$C-days criteria for $45^{\circ} <$ lat $< 85^{\circ}$ in the NH, but extended it

also to $35^{\circ} <$ lat $< 65^{\circ}$ in the SH. In all other cases and where the $5\,^{\circ}$C-days criteria fails, we still use the SMOKE-BEIS parameterization. The described algorithm written in python 2.7 has been included as Supplement S.5. An example map of the computed GLEN using the $5\,^{\circ}$C-days criteria in both hemispheres is shown in Fig. 3.

### 2.3.2 Photosynthetic photon flux density

From OpenIFS an accumulated surface PAR is available which is already integrated presumably over the spectrum $400 -$

$700\,\mathrm{nm}$. For this reason, it will be refered to as PPFD instead of PAR. This PPFD field has not been de-accumulated in the post-processing of the OpenIFS output and it is not feasible to redo all of the post-processing for this work. But for practical use in Eq. (20), this field needs to be de-accumulated. The main obstacle is that PAR has been accumulated since model start, so that the first field kept from the original OpenIFS simulation ($00\,\mathrm{UTC}$) is 12 hours after model start ($12\,\mathrm{UTC}$ on the previous day). In other words, the first time step of each day in the Oslo CTM3 has already accumulated PAR from $12\,\mathrm{UTC}$ on the



previous day. De-accumulation of times $03\,\text{UTC}$ to $21\,\text{UTC}$, simply means computing the difference

$$\text{PPFD}(t_i) = \text{PAR}(t_{i+1}) - \text{PAR}(t_i). \tag{35}$$

For de-accumulation of the remaining timestep, the best choice is subtracting the difference between $21\,\text{UTC}$ and $12\,\text{UTC}$ of the previous day

$$\text{PPFD}(t = 00\,\text{UTC}) = \text{PAR}(t = 00\,\text{UTC}) - [\text{PAR}(t = 21\,\text{UTC} - 1\,\text{day}) - \text{PAR}(t = 12\,\text{UTC} - 1\,\text{day})] \tag{36}$$

and limit the result to positive values only. An example PAR de-accumulation for January 2nd 2005 is shown in Supplement S.6 (Figs. S5–S7). The resulting PPFD fields are still accumulated over a time period of 3 hours and should be divided by 3. A known issue in the OpenIFS (cycles $\leq$ c41r2) causes surface PAR values to be about $30\,\%$ below observations. To counter this, we decided to refrain from the division at this stage, but need to bear this in mind for later OpenIFS cycles.

## 3 Evaluation

In this section, we present results from a manifold of Oslo CTM3 model integrations testing different parameters of the EMEP scheme. We focus on changes in ozone related to the stomatal conductance parameterization, e.g., total dry deposition $\sum \text{O}_3^{\text{DD}}$, dry deposition velocities $v_{\text{DD}}^{\text{O}_3}$, and surface concentration $[\text{O}_3](z_0)$ or surface burden $\text{O}_3(z_0)$ in units of $\text{kg}$, respectively. We evaluate our results with respect to the multi-model comparison of ozone dry deposition by Hardacre et al. (2015) (Section 3.2)

and observations (Section 3.3). The Oslo CTM3 is driven by meteorological input fields from ECMWF – OpenIFS cy38r1. CEDS historical emission inventory is used for anthropogenic emissions, while biomass burning is covered in daily resolution by NASA's Global Fire Emissions Database, Version4 (GFEDv4). Biogenic emissions are taken from MEGAN-MACC output (Sindelarova et al., 2014). In the following (Section 3.1), we will present the various model sensitivity studies.

### 3.1 Sensitivity studies

Due to significant differences between the EMEP scheme and the previous Wesely scheme with respect to implementation, it is not possible to fully disentangle and trace back every difference in results to a respective change. Therefore, we conducted in total nine sensitivity studies that mainly assess the parameter space of the stomatal conductance within the EMEP scheme. A reference simulation featuring the Wesely scheme has been conducted and will be referred to as *Wesely_type*, indicating that other implementations of the original work by Wesely (1989) may exist in other models. For all model integrations, the

meteorological reference year is 2005. This choice only affects the comparison with data and multi-model studies that either perform analysis on decadal averages or differing years. After finishing the work on the integration of the EMEP parameterization and initial testing, *EMEP_full* is the baseline simulation for the sensitivity studies, *EMEP_offLight* and *EMEP_offPhen*. As indicated by the names, these scenarios are extreme cases, setting $f_{\text{light}}$ and $f_{\text{phen}}$ to 1, respectively. During analysis of the results, we found, that the SMOKE-BEIS parameterization of the greening season did not extend to the boreal and subarctic re-

gions. Especially in Europe, this basically results in a suppression of canopy resistance in northern Scandinavia. The following





**Table 2.** Summary of specifications of all simulations discussed in this section. For simplicity, only the tested parameters are listed. An x denotes that the model was run exactly in the configuration as has been described in Section 2.

| Simulation | Wesely scheme | EMEP scheme | | | Greening season | | $v_{\text{ice/snow}}^{O_3}$ | Emissions |
| | | $f_{\text{phen}}$ | $f_{\text{light}}$ | $f_{\text{SW}}$ | SMOKE-BEIS | $5\,^\circ$C-days | (m s$^{-1}$) | (year) |
| --- | --- | --- | --- | --- | --- | --- | --- | --- |
| Wesely_type | x | | n/a | | x | n/a | 1/2000 | 2014 |
| EMEP_full | | incl. TR | x | SWLV1 | x | | 1/2000 | 2014 |
| EMEP_offLight | | incl. TR | 1 | SWLV1 | x | | 1/2000 | 2014 |
| EMEP_offPhen | | 1 | x | SWLV1 | x | | 1/2000 | 2014 |
| EMEP_SWVL4 | | incl. TR | x | SWLV4 | x | | 1/2000 | 2014 |
| EMEP_ppgs | | incl. TR | x | SWLV1 | | NH | 1/2000 | 2014 |
| EMEP_ppgs_2005 | | incl. TR | x | SWLV1 | | NH | 1/2000 | 2005 |
| EMEP_ppgssh | | excl. TR | x | SWLV1 | | NH+SH | 1/2000 | 2014 |
| EMEP_ppgssh_ice | | excl. TR | x | SWLV1 | | NH+SH | 1/10000 | 2014 |

sensitivity study therefore comprises two different versions of pre-processed greening season (refer to Section 2.3 for details): Mid and high latitudes in the northern hemisphere only (*EMEP_ppgs*) and for both, northern and southern hemisphere mid and high latitudes (*EMEP_ppgssh*). Building on the latter, we want to confirm the importance of the $v_{\text{ice/snow}}^{O_3}$ update (Helmig et al., 2007) in the Subarctic and Arctics, within the framework of the Oslo CTM3 (*EMEP_ppgssh_ice*). Accidentally, we have used

emissions for the year 2014 instead of 2005. Though, we take this as opportunity to assess the impact on ozone regarding differing emissions (*EMEP_ppgs_2005*). As mentioned previously, the simulations were initially conducted with the uppermost soil water level (SWVL1) in the implementation of $f_{\text{SW}}$. A final sensitivity study was conducted changing to soil water level at depths deeper than $1\,\text{m}$ (*EMEP_SWVL4*). All simulations discussed in this section are summarized in Table 2. An x in the table denotes that the model was run exactly in the configuration as has been described in Section 2.

In Fig. 4, we show the average relative difference between *EMEP_ppgsssh* and *Wesely_type* on global maps by means of surface ozone burden, dry deposition velocity and total ozone dry deposition. Surface ozone increases globally except for regions covered by tropical forest. Especially in desert regions in Africa and Asia, surface ozone increases by up to $100\,\%$. Consistently, dry deposition velocities decrease globally by the same order of magnitude in these regions, while they increase over tropical forest. With respect to total dry deposition, the picture is a bit less clear. We find a decrease of total dry deposition

of ozone in desert regions and ocean covered areas and an increase in regions covered by tropical forest, while at mid and high latitudes in both hemispheres only small changes are visible. A possible explanation to this divergence especially in desert regions is the difference between the prescribed dry deposition velocities in the Wesely scheme in comparison to those used in the EMEP scheme. We cover this in more detail in the following section.

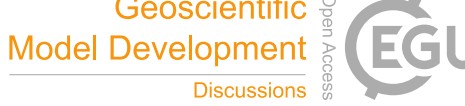



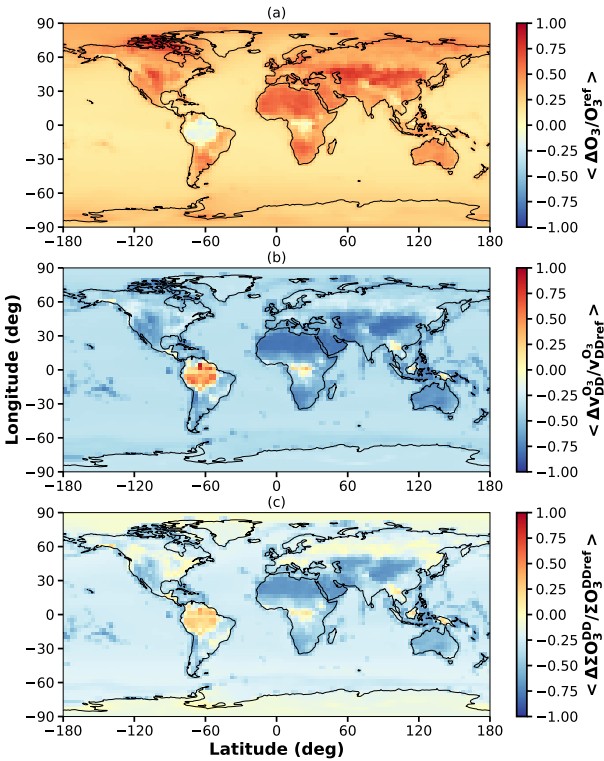

**Figure 4.** Relative difference between the sensitivity simulation *EMEP_ppgsssh* and the *Wesely_type* in (a) total surface ozone; (b) average ozone dry deposition velocity; (c) total amount of ozone removed from the atmosphere by dry deposition.

## 3.2 Comparison with modeling results

In our evaluation of results, we closely follow suggestions by Hardacre et al. (2015). For the purpose of comparison with the multi-model mean of the therein participating Task Force on Hemispheric Transport of Air Pollution (TF HTAP) models, we also have re-gridded our data to a horizontal resolution of $3° \times 3°$. In Section 3.2.1, we look at zonal distributions of $[O_3](z_0)$,

5   $v_{DD}^{O_3}$, and $\sum O_3^{DD}$ for all our sensitivity simulations and study seasonal cycles of hemispheric ozone as well as for nine land surface types which have been retrospectively separated (Section 3.2.2). From this, we estimate the total annual ozone dry deposition onto ocean, ice, and land surfaces and compare also with results from Luhar et al. (2017).

As dry deposition velocities are not directly available from the model output, monthly average dry deposition velocities $v_{DD}^{O_3}$ have been reconstructed from the ratio between the total ozone dry deposition $\sum O_3^{DD}(z_0)$ and monthly averaged surface ozone

10   amount $O_3(z_0)$

$$v_{DD}^{O_3} = \frac{\sum O_3^{DD}(z_0)}{O_3(z_0)} \cdot c_{\text{month}}. \tag{37}$$





Herein, $c_{\mathrm{month}} = \frac{\Delta h_{\mathrm{month}}}{s_{\mathrm{month}}}$, with the monthly average height of the lowermost model level in each gridbox $\Delta h_{\mathrm{month}}$ and the respective number of seconds in a month $s_{\mathrm{month}}$.

### 3.2.1 Zonal distribution

The annual zonal average with respect to surface ozone concentration (Fig. 5a displays on average, in consistency with Fig. 4a, a global increase of surface ozone concentrations by up to $10\,\mathrm{ppb}$ when applying the EMEP scheme compared to the Wesely scheme. This increase is largest in the zonal band $(25-50)\,^{\circ}\mathrm{N}$ which contains the major deserts. In the deep tropics $(5\,^{\circ}\mathrm{S} - 5\,^{\circ}\mathrm{N})$, the increase is smallest $(\Delta[\mathrm{O_3}] \leq 5\,\mathrm{ppb})$. We find that the EMEP scheme further intensifies the strong asymmetry between northern and southern hemisphere as a consequence of the distribution of the continental land masses and vegetation thereon. Among the sensitivity studies, there is only a low absolute variance. Most remarkable, but expected due to the much smaller prescribed dry deposition velocity over ice and snow, *EMEP_ppgssh_ice* displays almost a doubling of surface ozone in the high Arctics compared to *Wesely_type* but affects ozone concentrations down to latitudes at about $50\,^{\circ}$ in both hemispheres. Neglecting any dependence on solar radiation in the stomatal conductance formulation (*EMEP_offLight*) – or in other words allowing photosynthesis 24/7 – generally decreases the ozone concentration by about $2\,\mathrm{ppb}$, while choosing soil water at deeper levels than the surface layer (*EMEP_SWVL4*) increases $[\mathrm{O_3}](z_0)$ slightly. Rather surprisingly, switching off the phenology completely (*EMEP_offPhen*), amounts only to small difference (at most $3.5\,\%$ increase in the deep tropics). Switching from the SMOKE-BEIS parameterization to the pre-processed greening season (*EMEP_ppgs* compared to *EMEP_full*), surface ozone increases on average by $6\,\%$ between $(54-70)\,^{\circ}\mathrm{N}$. The scenario of differing emissions (2005 in comparison to 2014 or more specifically *EMEP_ppgs_2005* compared to *EMEP_ppgs*), yields higher ozone concentrations in the northern hemisphere in 2005 in accordance to a reduction in sulfur and $\mathrm{NO_x}$ emissions in south east Asia in later years. An opposite tendency is seen for latitudes south of $30\,^{\circ}\mathrm{N}$, probably owing to the industrial development of countries in these regions.

The $v_{\mathrm{DD}}^{\mathrm{O_3}}$ are shown in Fig. 5b. The dry deposition velocities in the EMEP scheme are on average $43\,\%$ lower than in the Wesely scheme. In the Arctics, except for *EMEP_ppgssh_ice*, all realizations of the EMEP scheme are constantly above the multi-model-mean. This indicates, that with respect to other models, the updated dry deposition velocity above ice and snow should be considered as new standard for the Oslo CTM3. This may, however, lead to an overcompensation of the current low-bias in surface ozone in the Oslo CTM3. The dry deposition velocities are of course independent of the emission scenario, but display a strong dependence on $f_{\mathrm{light}}$ and $f_{\mathrm{phen}}$. The closer to 1 these factors are the higher the dry deposition velocities become (compare *EMEP_offLight*, *EMEP_offPhen*, and *EMEP_ppgssh*). That highlights the importance of a proper PAR from meteorological input in CTMs or radiative transfer in PAR in global climate models (GCMs) and a proper choice of phenology for canopy conductance computation. Albeit, the total numbers are smaller, the shape of the normalized zonal average dry deposition velocities of the EMEP scheme and the multi-model-mean are similar (Supplement S.7: Fig. S8). The biggest exceptions are the zonal bands $(50-70)\,^{\circ}\mathrm{S}$, which is almost entirely covered by ocean, $(12-30)\,^{\circ}\mathrm{S}$, which coincides with the location of Australia and its desert regions, as well as its counterpart in the northern hemisphere $(12-30)\,^{\circ}\mathrm{N}$. However, it is not clear whether diverging dry deposition velocities over ocean or deserts have the larger impact.



The annual total ozone dry deposition is shown in Fig. 5c. In accordance to the previously described features, we observe a reduction of the global total ozone dry deposition in all sensitivity studies to almost one-half of the amount given by *Wesely_type*. The occurrence of this reduction in the zonal bands, where the major deserts are located, points to a change in $v_{\mathrm{desert}}^{\mathrm{O_3}}$. Consulting the parameter file used in the Wesely scheme, we indeed find $v_{\mathrm{desert}}^{\mathrm{O_3}} \equiv v_{\mathrm{tundra}}^{\mathrm{O_3}} = 0.26 \, \mathrm{cm\,s^{-1}}$ (Hough, 1991), while

in the EMEP scheme $v_{\mathrm{desert}}^{\mathrm{O_3}} = 0.05 \, \mathrm{cm\,s^{-1}}$ and $v_{\mathrm{tundra}}^{\mathrm{O_3}} = 0.24 \, \mathrm{cm\,s^{-1}}$, respectively. Similarly, dry deposition velocities over ice and snow and ocean have been even higher in the Wesely scheme ($v_{\mathrm{ice/snow}}^{\mathrm{O_3}} \equiv v_{\mathrm{water}}^{\mathrm{O_3}} = 0.07 \, \mathrm{cm\,s^{-1}}$) than in the original EMEP scheme ($v_{\mathrm{ice/snow}}^{\mathrm{O_3}} \equiv v_{\mathrm{water}}^{\mathrm{O_3}} = 0.05 \, \mathrm{cm\,s^{-1}}$). These differences in baseline dry deposition velocities over huge parts of the unvegetated surface of the Earth amount to most of the quantitative difference between the Wesely and the EMEP scheme. We further elaborate on this in the following section (Section 3.2.2).

There seems to be a discrepancy between the Oslo CTM3 response and the multi-model-mean, since the Wesely scheme is similar to the multi-model-mean with respect to total annual ozone dry deposition, while the reconstructed $v_{\mathrm{DD}}^{\mathrm{O_3}}$ of the EMEP scheme match better. This could be a sign of differences in photo-chemistry and transport (e.g. convective, advective, STE) between the Oslo CTM3 and the average TF HTAP model, but without comparing to the actual $[\mathrm{O_3}(z_0)]$ of the TF HTAP models that participated in the model intercomparison, we cannot elaborate on this any further.

In Appendix Fig. A1a, the average zonal ozone dry deposition is shown separated by month. Where available, we have added the multi-model-mean given by Hardacre et al. (2015) as reference. There are two major phases apparent: NH and SH greening season. Spring and summer in the NH is reflected in a pronounced peak of $v_{\mathrm{DD}}^{\mathrm{O_3}}$ in the northern mid latitudes, while it is absent in winter (SH summer). Spring and summer in the SH are marked by a southward shift of the tropical peak dry deposition velocity and a slight increase of $v_{\mathrm{DD}}^{\mathrm{O_3}}$ in the region $(20-40)\,^{\circ}\mathrm{S}$. In the Wesely scheme, NH mid latitude peak velocities appear in June

compared to July in the EMEP scheme, indicating that the seasonal cycles differ. The corresponding total monthly ozone dry deposition is shown in Appendix Fig. A1b. In general, the seasonal patterns are quite similar in the Wesely scheme and the EMEP scheme, displaying a strong symmetry around $10\,^{\circ}\mathrm{N}$ in January/February and November/December, respectively. What differs most is the molding and intensity of the NH peak dry deposition. Both schemes reach the maximum in June/July but the peak is much more differentiated in March already in the Wesely scheme. Similarly, the SH tropical peak dry deposition

is reached in August/September but sustained longer, into October, in the Wesely scheme. Since we have not conducted any simulation with a meteorological year other than 2005, we cannot elaborate on whether this is a special feature of our chosen year or not. In comparison, the available 2 months of the multi-model-mean display a somewhat different pattern. Especially the NH/SH asymmetry is much more pronounced in February, than it is the EMEP scheme.

### 3.2.2 Average seasonal cycles

To further disentangle the contributions of different regions to the global ozone budget, we will briefly look at different projections of seasonal cycles.

In Fig. 6, the total annual ozone dry deposition separated into mid and high latitudes in the northern hemisphere $(30^{\circ}-90^{\circ}\mathrm{N})$, the tropics and subtropics $(30^{\circ}\mathrm{S}-30^{\circ}\mathrm{N})$, and the mid and high latitudes in the southern hemisphere $(30^{\circ}-90^{\circ}\mathrm{S})$ is shown. We have added the multi-model-mean by Hardacre et al. (2015) as reference. As expected, the NH mid and high





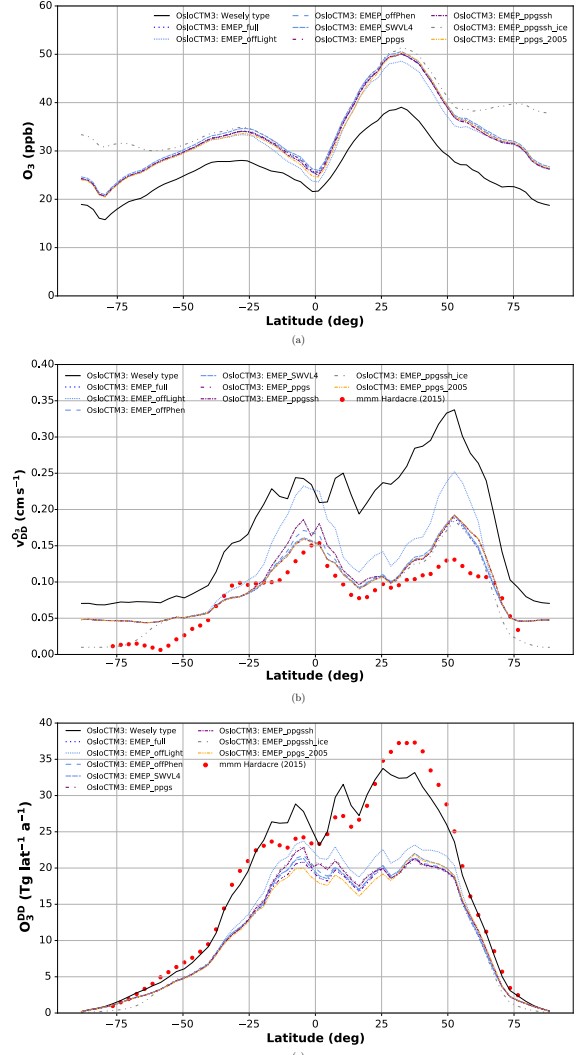

**Figure 5.** Comparison of the manifold of Oslo CTM3 integrations with respect to (a) Surface ozone concentrations, (b) Annual average ozone dry deposition velocity, (c) Total annual ozone dry deposition. The different colors indicate sets of simulation with similar baselines. The multi-model mean from the evaluation of TF HTAP models by Hardacre et al. (2015) is shown as a reference (where available).

latitudes display a strongly pronounced seasonal cycle, while it is less pronounced in the tropics (due to the lack of seasons) and in the SH (due to the small percentage of vegetated surface). The highest ozone dry deposition is found in the tropics and amounts on average to the peak level of dry deposition in the NH for the multi-model-mean (Hardacre et al., 2015) and EMEP scheme. In the Wesely scheme, the average tropical ozone dry deposition diverges by about $5\,\mathrm{Tg}$ in comparison to its

5    corresponding NH maximum. The seasonal cycle in the EMEP scheme appears to be shifted towards later in the year in the





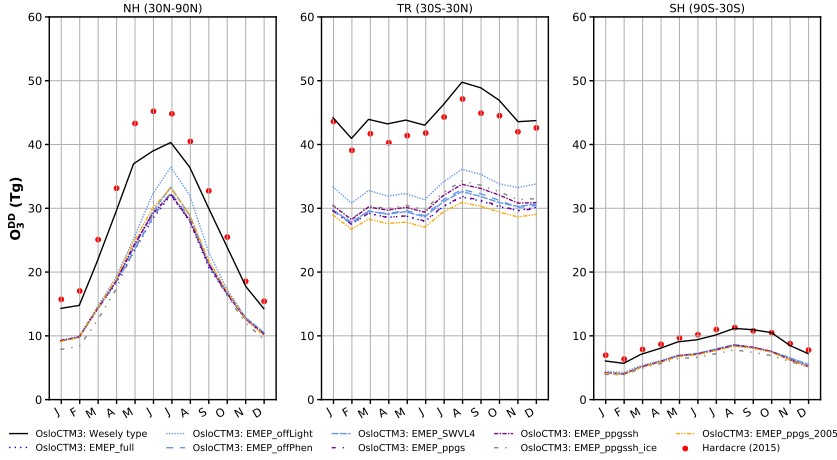

**Figure 6.** Seasonal cycle of total annual amount of ozone removed from the atmosphere through dry deposition separated into northern hemisphere (NH), tropics (TR), and southern hemisphere (SH). The multi-model mean from the evaluation of HTAP models by Hardacre et al. (2015) is shown as a reference.

NH. The seasonal cycle in the tropics and subtropics is not changing, but the total amount of dry deposition of ozone is very sensitive to differing realizations of the EMEP scheme, with *EMEP_offLight* and *EMEP_ppgs_2005* displaying the highest and the lowest amount, respectively. This indicates that surface ozone in the tropics may be very sensitive to changes in emissions of ozone precursors and changes in long-wave radiative transfer (e.g., due changes in cloud cover or aerosol).

As suggested by Hardacre et al. (2015), we retrospectively separate ozone dry deposition velocities with respect to surface types. Based on a CLM 2 average dynamic land surface map, we generated masks for 9 different surface types (Fig. B1a and used these to select gridboxes with a high percentage of these surface types, ranging from a meager $70\%$ for cropland in the NH mid latitudes to $100\%$ regarding desert, ocean, snow and ice, and tropical forest. Thus, it was not possible to exclusively select gridboxes with $100\%$ cover for each surface type. Since we have not performed a full unfolding on the

data, the results should be treated with slight caution. In Fig. 7, the seasonal cycles of dry deposition velocities are shown for the nine surface categories. The patterns and absolute numbers differ substantially between the Wesely scheme and the EMEP scheme and the multi-model-mean. The divergence of the average dry deposition velocities between the Wesely scheme and the EMEP scheme in desert regions ($\Delta\overline{v}_{\text{desert}}^{\text{O3}} = 0.18\,\text{cm}\,\text{s}^{-1}$) as well as grassland ($\Delta\overline{v}_{\text{grassland}}^{\text{O3}} = 0.54\,\text{cm}\,\text{s}^{-1}$) is quite remarkable as well as the difference of both to the multi-model-mean in tropical forest regions ($\Delta\overline{v}_{\text{tropical forest}}^{\text{O3}} = 0.49\,\text{cm}\,\text{s}^{-1}$). In contrast

to the Wesely scheme and the EMEP scheme, the multi-model-mean indicates a rather pronounced seasonal cycle in desert regions ($0.10\,\text{cm}\,\text{s}^{-1} \leq v_{\text{desert}}^{\text{O3}} \leq 0.15\,\text{cm}\,\text{s}^{-1}$). The estimated dry deposition velocities over desert regions are consistent with the previously mentioned baseline values of ozone dry deposition, which means that in the EMEP scheme they are about 1 order of magnitude lower than in the Wesely scheme. From a limited number of ozone flux measurements in the Sahara desert, Güsten et al. (1996) deduced $v_{\text{desert, day}}^{\text{O3}} = 0.1\,\text{cm}\,\text{s}^{-1}$, $v_{\text{desert, night}}^{\text{O3}} = 0.04\,\text{cm}\,\text{s}^{-1}$, and $\overline{v}_{\text{desert}}^{\text{O3}} = 0.065\,\text{cm}\,\text{s}^{-1}$. This implies, that ozone dry



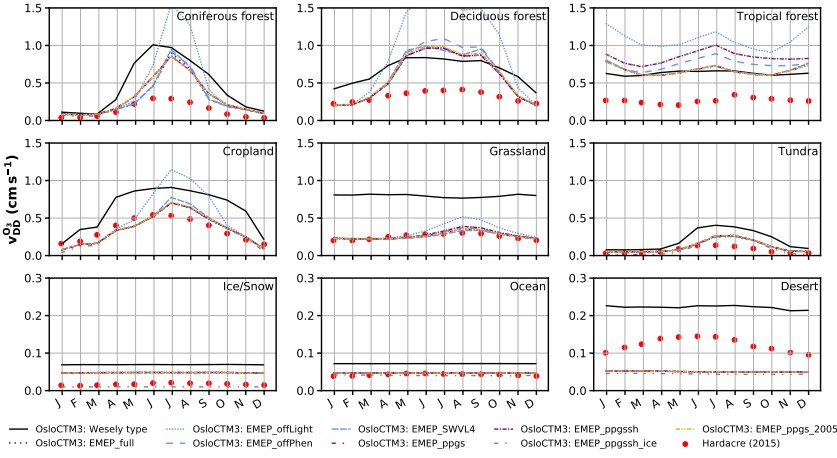

**Figure 7.** Average seasonal cycles of ozone dry deposition velocities separated by land use type. Results from (Hardacre et al., 2015) are shown as a reference. We refrain from showing the full extent of *EMEP_offLight* here, since it is an extreme scenario and has been discussed already.

deposition over desert regions is highly overestimated in the Wesely scheme as well as in TF HTAP models, while it may be underestimated in the EMEP scheme. Similarly, the dry deposition velocities over water differ. From measurements during ship-campaigns a mean value of $\overline{v}_{water}^{O_3} = 0.019\,\mathrm{cm\,s^{-1}}$ over the ocean has been deduced (Helmig et al., 2012). In a model study of different mechanisms of dry deposition to ocean waters by means of prescribed $\overline{v}_{water}^{O_3}$ and one- and two-layer gas exchange

modeling, Luhar et al. (2017) found $\overline{v}_{water}^{O_3}$ ranging between $0.018\,\mathrm{cm\,s^{-1}}$ (two-layer scheme) and $0.039\,\mathrm{cm\,s^{-1}}$ (prescribed). With $\overline{v}_{water}^{O_3} = (0.046 \pm 0.002)\,\mathrm{cm\,s^{-1}}$, the EMEP scheme (Section 2.1.2) yields probably a too strong dry deposition to ocean which implies that ozone concentrations might even become larger and dry deposition even lower in the model if a two-layer scheme would be implemented. With respect to vegetation, we might be able to improve the model performance further by allowing more PFTs and phenologies, especially in regions covered by tropical or coniferous forest (Anav et al., 2017).

Finally, we take a look at the different global as well as hemispheric dry deposition sinks for ozone (Table 3). Despite its vastness, the ocean amounts only to about $23\,\%$ of the global ozone sink due to dry deposition in the Oslo CTM3 with operative EMEP scheme, while ice and snow account for far less than $1\,\%$. The remainder is deposited to various other land surfaces of which deserts might be the most neglected in process-modeling. The total annual dry deposition in the EMEP scheme is about one-third below the multi-model-mean result by Hardacre et al. (2015). But also the results of Luhar et al. (2017) yield a

$(19 - 27)\,\%$ lower ozone dry deposition than the models participating in the model intercomparison, with deposition to ocean ranging between $(12 - 21)\,\%$.





**Table 3.** Total ozone dry deposition for the respective year in $\mathrm{Tg\,a^{-1}}$ separated into ocean, ice and, land contributions.

| Simulation | Ocean | | | Ice | | | Land | | | Total | | | $\Delta^{\dagger}$ |
| | NH | SH | Global | NH | SH | Global | NH | SH | Global | NH | SH | Global | |
| | | $(\mathrm{Tg\,a^{-1}})$ | | | $(\mathrm{Tg\,a^{-1}})$ | | | $(\mathrm{Tg\,a^{-1}})$ | | | $(\mathrm{Tg\,a^{-1}})$ | | (%) |
|---|---|---|---|---|---|---|---|---|---|---|---|---|---|
| Wesely_type | 88.7 | 110.3 | 199.0 | 0.7 | 5.0 | 5.7 | 346.1 | 153.5 | 499.6 | 612.4 | 347.9 | 960.2 | 46.8 |
| EMEP_full | 67.1 | 84.7 | 151.8 | 0.6 | 4.4 | 5.0 | 236.0 | 112.3 | 348.2 | 408.6 | 245.6 | 654.2 | 0.0 |
| EMEP_offLight | 66.6 | 84.1 | 150.7 | 0.6 | 4.4 | 5.0 | 267.8 | 129.8 | 397.6 | 451.3 | 267.2 | 718.6 | 9.8 |
| EMEP_offPhen | 67.1 | 84.6 | 151.7 | 0.6 | 4.4 | 5.0 | 239.9 | 116.0 | 355.9 | 413.9 | 249.9 | 663.7 | 1.5 |
| EMEP_SWVL4 | 68.1 | 86.1 | 154.2 | 0.6 | 4.5 | 5.1 | 241.8 | 115.0 | 356.8 | 417.2 | 250.7 | 667.9 | 2.1 |
| EMEP_ppgs | 67.1 | 84.6 | 151.7 | 0.6 | 4.4 | 5.0 | 238.7 | 112.6 | 351.3 | 411.4 | 246.0 | 657.4 | 0.5 |
| EMEP_ppgssh | 67.0 | 84.5 | 151.6 | 0.6 | 4.4 | 5.0 | 243.9 | 119.4 | 363.3 | 419.4 | 253.6 | 673.0 | 2.9 |
| EMEP_ppgssh_ice | 66.5 | 84.8 | 151.3 | 0.2 | 1.3 | 1.4 | 240.3 | 120.4 | 360.7 | 413.0 | 251.1 | 664.1 | 1.5 |
| EMEP_ppgs_2005 | 66.8 | 83.5 | 150.4 | 0.6 | 4.4 | 5.0 | 237.8 | 110.0 | 347.8 | 409.2 | 240.7 | 649.9 | -0.7 |

$^{\dagger}$: Difference of global annual total relative to *EMEP_full*.

### 3.3 Comparison with observations

In this section, we compare our results to observations. For all comparisons, we use the original resolution of the Oslo CTM3 ($2.25° \times 2.25°$) instead of the re-gridded ($3° \times 3°$).

In Fig. 8a, seasonal cycles of average ozone dry deposition fluxes for six observation sites are shown. We have computed 5 a model average for all sensitivity studies at the closest grid point and show the $1\sigma$ error band. The shaded area around the multi-model-mean indicates the broad range of model results but is not an actual error band since such is not given in Hardacre et al. (2015). At about $4$ of $6$ sites, the EMEP scheme performs better than the Wesely scheme and better than or the same as the multi-model-mean. We use a $\chi^2$-test

$$\chi^2 = \sum_{i=1}^{12} \frac{\left(\overline{\mathrm{O_{3DD,}}_i}^{\mathrm{sim}} - \overline{\mathrm{O_{3DD,}}_i}^{\mathrm{obs}}\right)^2}{\sigma_i^2}, \tag{38}$$

10 with an estimated standard deviation of observation $\sigma_i = 1\,\mathrm{mmol\,m^{-2}\,s^{-1}}$ and divide it by the number of degrees of freedom (NDF) to assess this subjective analysis in a more objective way. The closer to $1$ this test scores, the better does the simulation represent the observation. A score between $0$ and $1$ indicates that the estimated $\sigma$ is too small. The results of the $\chi^2$-test are shown together with the divergences in Fig. 8b. The $\chi^2$-test reveals also that in $4$ of $6$ cases the EMEP scheme improves the performance of the Oslo CTM3 with respect to ozone dry deposition fluxes, although a satisfying result is only achieved for two 15 sites (Castel Porziano, Blodgett Forest). With only one full year of simulation, the model uncertainty regarding the seasonal cycle at observational sites cannot be properly quantified. Furthermore, the observational averages comprise at most $9$ years worth of data. Statistically, these data may still subject to interannual variability. Among other aspects, the horizontal as well





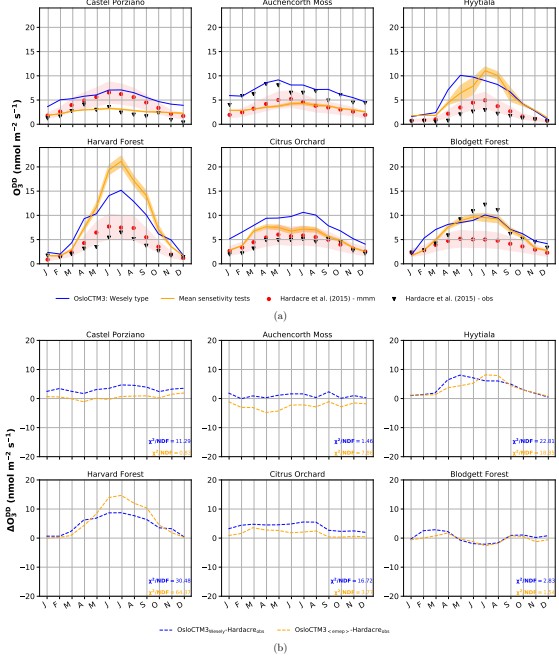

**Figure 8.** Ozone dry deposition fluxes at different observation sites. (a) Comparison between Wesely scheme and average result from all sensitivity studies with observational averages taken from Hardacre et al. (2015). The model uncertainty of the Oslo CTM3 is given as $1\sigma$ error band. The shaded area around the multi-model-mean indicates the broad range of different model results but is not an actual error band. (b) Model divergence from observation and $\chi^2$-test results.

as vertical resolution play an important role in the model performance. Although, we do not explicitly assess the impacts of differing resolutions in our model, we can assume that both high and low biases exist due to dilution of sources and sinks in coarse resolution models (Schaap et al., 2015). Good matches between observation and model are only to be expected if the station's location is representative for an area similar to the respective model gridbox.

## 4  Discussion and conclusions

We have presented an update of the dry deposition scheme in the Oslo CTM3 from purely prescribed dry deposition velocities (Wesely, 1989; Hough, 1991) to a more process-oriented parameterization taking the state of the atmosphere and vegetation into account (Simpson et al., 2012). In our implementation based on Simpson et al. (2012), we follow a classical, resistance analogous approach in computing contributions to dry deposition. The quasi-laminar resistance in the dry deposition scheme is capable of adjusting to surface wind stress but falls back to its prescribed limits in most of the tested cases. The surface resistance computation is divided into stomatal and non-stomatal resistance. The stomatal resistance is sensitive to parameters




associated with the vegetation such as photosynthetic active radiation, water vapor pressure deficit, soil water content, and plant phenology, while the non-stomatal resistance is affected by the existence or absence of vegetation.

The annual amount of ozone dry deposition decreases by up to $100\%$ changing from the old dry deposition scheme to the new one. Compared to results from a multi-model evaluation (Hardacre et al., 2015), the total annual ozone dry deposition
in the Oslo CTM3 is about $33\%$ below average. However, there seems to be a tendency that newer TF HTAP models show a lower total annual dry deposition of ozone than older models, indicating that newer developments lead to decreasing ozone dry deposition and increasing tropospheric ozone burden (e.g., Luhar et al., 2017; Hu et al., 2017).

We found the response of the Oslo CTM3 to the changes in dry deposition velocities from the old and the new dry deposition scheme to be consistent. A decrease in $v_{DD}^{O_3}$ leads to a decrease in total ozone dry deposition and an increase in surface ozone
concentration $[O_3(z_0)]$. As the new scheme is more similar to the multi-model-mean (Hardacre et al., 2015) with respect to dry deposition velocities, while the old scheme agrees better in terms of total dry deposition, there is an apparent discrepancy when comparing with the multi-model-mean. Without knowing $[O_3(z_0)]$ of the TF HTAP models, this cannot be resolved and may hint to, e.g., differences in photo-chemistry, stratosphere-troposphere-exchange as well as to differences in aerosol loadings among many other aspects.

Most of the decrease in ozone dry deposition in the Oslo CTM3 can be attributed to changes in dry deposition velocities over the ocean and deserts. This is mainly due to updates of the respective prescribed dry deposition velocities $v_{DD}^{O_3}$. In case of desert and grasslands the difference between the old and new prescribed value is at the order of 1 magnitude. Over the ocean, the absolute change in dry deposition is small, but it is accumulated over a large area which is especially amplified in the southern hemisphere. Small adjustments to the lower limits in our quasi-laminar layer resistance formulation may help
improve the Oslo CTM3 performance in this regard. With respect to available measurements of dry deposition velocities of ozone over desert (Güsten et al., 1996) and ocean (Helmig et al., 2012), the new Oslo CTM3 dry deposition scheme slightly underestimates ozone dry deposition velocities over the former and overestimate them over the latter. Regarding the vastness of the ocean and the ongoing desertification, it may be worthwhile to study dry deposition in these regimes at a more process-oriented level, e.g., 2-layer gas exchange with ocean waters (Luhar et al., 2017), wave braking and spray (Pozzer et al., 2006).

Although dry deposition to ice and snow amounts to less than $1\%$ of the total global annual ozone dry deposition, a decrease in prescribed dry deposition velocity in accordance to combined measurements and model studies (Helmig et al., 2007) causes almost a doubling in the surface ozone concentrations in the high Arctics and affects surface ozone concentrations down to latitudes at about $50°$ in both hemispheres. By comparing with results from the multi-model evaluation (Hardacre et al., 2015), we conclude that it is important to use this updated ozone dry deposition velocity to counter an Arctic surface ozone low-bias
in the model, however, this could lead to an overcompensation (high-bias).

We have studied the parameter space of the stomatal conductance parameterization and found that total surface ozone in the tropics is most sensitive to changes in ozone uptake by plants. The difference between our most extreme test cases amounts to $15\%$ with respect to total surface ozone, while the more realistic test case using differing years of emission amounts to about $5\%$. This may indicate that future changes in vegetation cover and solar radiation at the surface due to changes in stratospheric
ozone, cloud cover, or aerosols could also strongly influence the surface ozone burden in the tropics. Total column ozone in





the tropics is predicted to decrease due to changes in the atmospheric circulation (e.g., WMO - Global Ozone Research and Monitoring Project, 2014), while tropospheric and surface ozone increase. The combined effects of increasing emissions of ozone precursors and an increase in UV due to thinning of stratospheric ozone might permit more UV light at ground and thus increase the ozone production.

In the northern hemisphere mid and high latitudes $(50\,^\circ - 75\,^\circ\mathrm{N})$, total surface ozone increases by about $7.7\,\%$ if the beginning and end of the vegetation period is estimated based on a $5\,^\circ\mathrm{C}$-days criteria instead of prescribed. This is very important for any study focusing on ozone in the boreal and subarctic regions.

An important factor in the global ozone budget are emissions of precursor substances. We cover this by using the same meteorology with different years of CEDS emissions. We chose the years 2005 and 2014 for our comparison. In 2014, surface
ozone burden is higher in the southern hemisphere and in the tropics $(\sim 5\,\%)$ compared to 2005, while it is lower in the northern hemisphere $(\sim 2\,\%)$. This is most likely reflecting the ongoing industrialization process of countries in the southern hemisphere and the commitment and implementation of air quality regulations of industrialized nations in the northern hemisphere.

We also evaluated the model with respect to observed dry deposition velocities at six sites in the northern hemisphere and found that the updated dry deposition scheme performs better than the old one, but is not able to reproduce the measurements
at most sites quantitatively. This may be due to several reasons. The model resolution in both horizontal $(2.25^\circ \times 2.25^\circ)$ and vertical (L60, $P_{\max} = 0.02\,\mathrm{hPa}$) does not capture all details in transport, thus affecting the distribution and transport (e.g., long-range, convection, and stratosphere–troposphere exchange) of ozone and its precursors. Depending on the location of the observation site and its respective representativeness for a larger area, ozone dry deposition and ozone concentrations are expected to be over- or underestimated in the model. In addition, a comparison of very few years of measurement to only one
specific year of simulation may reflect the year to year variability more than the actual model performance.

Future work on the Oslo CTM3 should include a direct output of dry deposition velocities for diagnostic purposes. The model performance could also be improved by allowing for more plant functional types and phenologies than currently used or implementing an actual photosynthesis-based modeling of plants. The photolysis- and chemical reaction computation as well as reaction rates shall also be revised in the future.

*Code and data availability.*   The Oslo CTM3 is publicly available on git-hub under a MIT license. Model results can be made available under request.

**Appendix A: Figures**

*Author contributions.*   Stefanie Falk has compiled the manuscript, finalized the implementation of the stomatal conductance in the EMEP-based dry deposition scheme of the Oslo CTM3, conducted the simulations, and analysed and evaluated the results. Amund Søvde Haslerud





**Figure A1.** Comparison of the manifold of Oslo CTM3 integrations with respect to (a) Zonal average ozone dry deposition velocities; (b) Total annual amount of ozone removed from the atmosphere via dry deposition. The multi-model mean from the evaluation of TF HTAP models by Hardacre et al. (2015) is shown as a reference (where available).





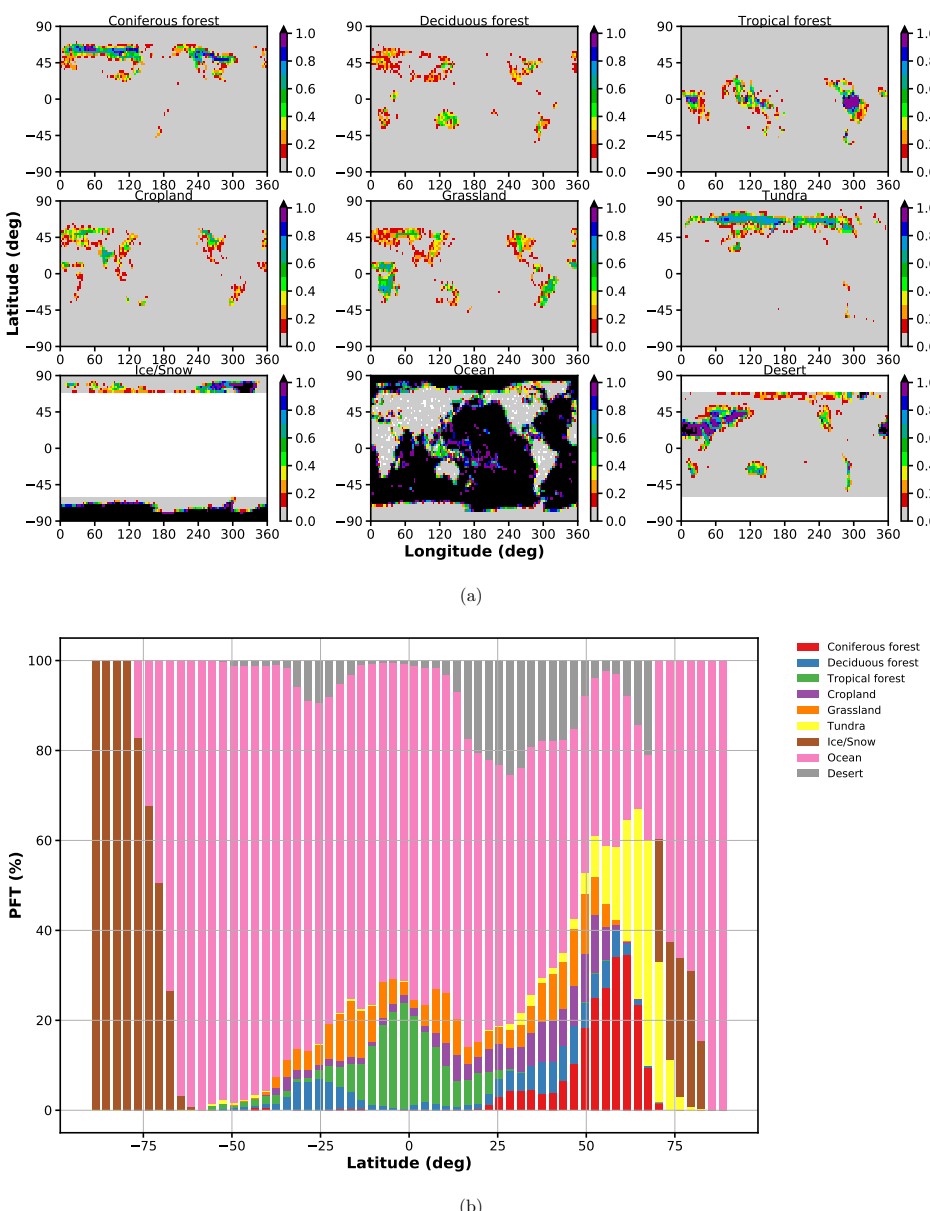

(a)

(b)

**Figure B1.** Partitioning of land surface types. (a) CLM 2 dynamic land surface types in $(0.5 \times 0.5)^\circ$ resolution; (b) Zonal distribution of land surface types.



has implemented the EMEP-based dry deposition scheme and wrote the respective documentation. Both authors contributed to the writing and discussion of the paper.

*Competing interests.* The authors declare that they have no conflict of interest.

*Acknowledgements.* This work was supported by the Norwegian Research Council (NRC) through the project The double punch: Ozone and
climate stresses on vegetation (268073).

The simulations were performed on resources provided by UNINETT Sigma2 – the National Infrastructure for High Performance Computing and Data Storage in Norway (project nn2806k).

The used Leaf Area Index (LAI) and roughness length ($z_0$) are available online from Oak Ridge National Laboratory Distributed Active Archive Center, Oak Ridge, Tennessee, U.S.A. (https://doi.org/10.3334/ORNLDAAC/970).

Community Emission Data System (CEDS) historical emission inventory is provided by the Joint Global Research Institute project (http://www.globalchange.umd.edu/ceds/.) Randerson, J.T., G.R. van der Werf, L. Giglio, G.J. Collatz, and P.S. Kasibhatla. 2018. Global Fire Emissions Database, Version 4, (GFEDv4). ORNL DAAC, Oak Ridge, Tennessee, USA. https://doi.org/10.3334/ORNLDAAC/1293 We would like to thank Prof. Frode Stordal (Section for Meteorology and Oceanography, University of Oslo) for discussions regarding early drafts of the manuscript as well as Anne Fouilloux (scientific programmer at the same institute) for technical support. We would also like to
thank the Center for International Climate Research (CICERO) for their support of this work.



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
