# Peer review of "Update and evaluation of the ozone dry deposition in the Oslo CTM3 v1.0"

_Geoscientific Model Development, 2019_

## Referee Comment (RC1) · Anonymous Referee #1 · 2 Apr 2019

This paper describes and evaluates an update of the ozone dry deposition scheme implemented in the Oslo CTM3 global model. The update involves a move to a more process-based parameterisation of the terms that particularly appear in the land component of the deposition scheme. The paper reads generally well. I have a number of comments which would require a major revision of the paper, and these are given below.

1. Last line in the abstract: "While high sensitivity to changes in dry deposition to vegetation is found in the tropics, the largest impact on global scales is associated to changes in dry deposition to the ocean and deserts." The authors do not provide details in the paper as to what has changed in the updated scheme for such an impact. Is it the surface resistance ($R_c$) value? Or the other two resistances ($R_a$ and $R_b$)? What

[Figure]

are the typical values?

What value of Rc for water has been used, and how it compares with the value used in the Wesely scheme?

2. Table 3: Why the deposition values for ocean + ice + land do to add up to the total values reported, for all simulations?

3. Table 3: The new land-based deposition values are much lower than what has been reported in previous studies (e.g. Hardacre et al., 2015) and the authors largely attribute this to the changes in the updated scheme for the desert surface type. However, the paper does not provide any observational support to back this up. Are there any relevant deposition measurements (velocity or flux) that can be used for this purpose? At least, some comparison with ozone measurements (or even O3 reanalyses) should be provided for this surface type (and perhaps others) to see if the model is heading in the right direction with the updated deposition scheme.

4. Table 3: It will also be useful to report the global ozone burden from the various simulations.

5. Section 2.1.1, Eq. (2): The statement "For certain values of z, z0, and L, this may result in nonphysical (negative) values for Ra." I do not comprehend as to why this would occur since this equation is simply based on the well-used Monin-Obukhov similarity theory (MOST) for the surface layer. This occurrence would also imply negative wind speeds. Actually Eq. (2) is incorrect: the term psi_m((z-d)/z0) should be psi_m((z-d)/L), and the sign of the third term on the right-hand side should be positive (not negative). Given that (z-d) > z0 (assuming the model is formulated correctly), Eq. (2) should always yield positive values.

Eqs. (3–5): I am not sure why Monteith (1973) needs to be invoked here. Given that the term in the square brackets on right hand side of Eq. (2) is equal to k.u(z)/u* as per MOST, substituting this into Eq. (2) results in Eq. (5).

Define z, z0 in Eq. (2). The parameter d is the so-called displacement height, and is not a constant (depends on the surface type).

6. Page 2, line 31: The first reference to the Oslo CTM3 in the body of the paper is made here as "...we have not implemented any parameterization of these processes in the Oslo CTM3 as of now." Some brief introductory text is required here (or better at the start of the paragraph) to introduce the model properly. Also, the text between lines 25 – 33 on what is not considered in the model is too detailed to be here, so shorten and move it to Section 2.

7. Page 3, line 19; page 3, line 28; page 21, last line: There is a newer ozone dry deposition study by Luhar et al. (2018, ACP, 18, 4329-4348) which, using global ozone reanalyses and a more realistic process-based oceanic deposition scheme, estimates the total global deposition at 722.8 ± 87.3 Tg O3 yr-1, which includes an oceanic component of 98.4 ± 30.0 Tg O3 yr-1. These figures should be cited for comparison.

8. Section 2.2: Since the present paper is about ozone dry deposition, this section seems like a distraction and hence should be omitted.

9. Page 14, lines 15 – 18: Anthropogenic, biomass burning, and biogenic emissions are included in the model. How are other emissions such as soil NOx, wetland methane, and oceanic methane and CO specified?

10. Page 15, line 4: The statement "Accidentally, we have used emissions for the year 2014 instead of 2005." It is not clear what the consequences on the results are of this?

11. Section 3.2.1: In the Fig 5 discussion, although snow and ice is discussed, there is no discussion on the oceanic differences between the present study and Hardacre et al. (2015). This is particularly important for the Southern latitudes.

12. Section 3.2.1: The Hardacre et al. (2015) simulations were for the year 2001, whereas the present study is mostly for the year 2015 emissions (see Table 1) driven by the year 2005 meteorology. In addition, the observational averages used in Fig. 8

are based on multi-year data. The authors should discuss the implications of these differences about different years on the deposition results presented (e.g. uncertainty).

13. Page 24, lines 3 – 4: "The annual amount of ozone dry deposition decreases by up to 100% changing from the old dry deposition scheme to the new one." Table 3 does not support this, but this may be true for some surface types. So please qualify the statement.

14. Page 24, line 15: "Most of the decrease in ozone dry deposition in the Oslo CTM3 can be attributed to changes in dry deposition velocities over the ocean and deserts." What are the dominant factors in these changes? For example, is it mostly the surface resistance (Rc) term? For the ocean, it is likely to be Rc. For deserts, maybe Rb? Is it possible to quantify these differences in the resistance terms?

15. Page 24, line 24: "2-layer gas exchange with ocean waters (Luhar et al., 2017)." As mentioned earlier, Luhar et al. (2018) has derived a more realistic process-based deposition scheme for the ocean, but the results for deposition velocity do not seem to be too different from those in Luhar et al. (2017).

16. Page 25, lines 11 – 12: The comment "This is most likely reflecting the ongoing industrialization process of countries in the southern hemisphere and the commitment and implementation of air quality regulations of industrialized nations in the northern hemisphere" is quite speculative and may be omitted.

17. Eq. (13) cf. Eq. (14): g_STO or G_sto – use consistency with notation.

18. The first half of the abstract, the text before "In this paper…," is introductory material and can be deleted.

19. In the abstract (lines 15-16), it is better to say "…leading to an increase in surface ozone of up to 100% in some regions."

20. Page 22, line 7: "At about 4 of 6 sites." About? Not sure?

---

## Referee Comment (RC2) · David Simpson (Referee) · 10 Apr 2019

**Review of Falk & Haslerud, Update and evaluation of the ozone dry deposition in the Oslo CTM3 v1.0**

David Simpson, EMEP MSC-W, April 2019

This paper summarises an update in the dry deposition framework of the Oslo CTM3 model. Many of the equations and ideas are taken from the EMEP model of Simpson et al. (2012) (which I will refer to as S2012 below), but the equations are applied in ways that seem to differ substantially from the S2012 approach. Although differences are expected, I have trouble following the logic behind the CTM3 formulation, and in some cases I would say it is incorrect.

I apologise for a somewhat harsh tone below, but I am afraid to say that I found too many difficulties with both the concepts and the clarity of this paper. I cannot recommend the paper for publication without major revision to both the paper and probably the underlying model construction.

**Major points**

**A) Mosaic formulation**

In the EMEP model, the deposition flux ($F^i_{g,k}$) of a gas $i$ to the ground surface over ecosystem $k$ is modelled using the so-called deposition velocity, $V^i_{g,k}(z)$, such that:

$$F^i_{g,k} = -V^i_{g,k}(z) \times \chi^i_k(z) \tag{1}$$

This equation is assumed to be true throughout the so-called constant flux layer. In S2012 we assume that the concentration and deposition velocity calculated at the centre of the lowest grid cell ($\sim 45$ m in S2012, a height we refer to below as the reference height $z_{ref}$) is within this layer, and thus we can set all $\chi^i_k(z_{ref})$ equal to one average value, $\chi^i_{avg}(z_{ref})$. The deposition velocities over each land-cover are calculated using standard big-leaf approaches as:

$$V^i_{g,k}(z) = \frac{1}{R_{a,k}(z) + R^i_{b,k} + R^i_{c,k}} \tag{2}$$

To calculate grid-average deposition fluxes, EMEP implements a so-called mosaic approach, whereby the the grid-average deposition rate is given by:

$$\overline{V^i_{g,k}(z)} = \sum_{k=1}^{N} f_k \times V^i_{g,k}(z) \tag{3}$$

where the overline symbolises the grid-square average, $f_k$ is the fraction of land-cover type $k$ in the grid-square, and $V_{g,k}^i$ is the deposition velocity for this land-cover type. An important point about the S2012 approach is that it calculates all terms (including $u_*$, $R_a$, etc.) consistently for each land-cover, and then averages the fluxes.

CTM3 claims to implement a mosaic approach, but instead of calculating deposition rates over each land-cover, and then aggregating using Eqn.3, CTM3 seems to perform the following steps:

$$R_a = \frac{u_z}{u_*^2} \tag{4}$$

$$\overline{Gs_g^i(z)} = \sum\nolimits_{k=1}^N f_k \times Gs_{g,k}^i(z) \tag{5}$$

$$\overline{Gns_g^i(z)} = \sum\nolimits_{k=1}^N f_k \times Gns_{g,k}^i(z) \tag{6}$$

As far as I can tell from the text, $R_a$ is calculated once, with the same value of $u_*$ for all land-covers. The CTM3 $R_b$ calculation seems to also use the same $u_*$ over different land-covers, except over sea where a more sophisticated scheme is used. Equations 5-6 above are equivalent to Eqns (22) and (26) from the manuscript. Now I am puzzled however as to how all this is put together. Do they use:

$$Gc = \text{LAI} \, \overline{Gs_g^i(z)} + \overline{Gns_g^i(z)} \tag{7}$$

as suggested by their Eqn (13) (though I added overlines here), followed by $R_c = 1/G_c$ and then the manuscript's Eqn (1) for deposition velocity? (In this case, which LAI is used?)

In any case, I think this approach has serious problems. Why average first for $G_s$ and then for $G_{ns}$, when it is the fluxes ($F_k$, or $V_{g,k}^i(z_{ref}) \times \chi_{avg}^i(z_{ref})$) which need to be averaged? I also do not understand why they would use the same $u_*$ and $R_a$ for all land-covers. I think the authors need to make a case for their approach, or change it.

**B) Calculation of $R_a$**

$R_a$ in CTM3 appears to be calculated just once, and from a height of 8 m. This means that there is no consistency between the $R_a$ term and the underlying surface, which is clearly wrong.

The similarity equation for $R_a$ given in their Eqn (2) is very standard and has been used for decades (Garratt 1992). As pointed out by Ref 1, the equation as given has errors. The correct equation will not give negative values unless presented with wrong inputs, and I suspect that that is what has happened. It is actually difficult to tell what

was tested from the manuscript though, since they state simply that $d$ is 'typically 0.7 m'. Did they use $d$ properly, consistent with the underlying land-cover and its $z_0$? Did they assume that their 8 m meteorology was at a physical height of 8 m, or at $d + 8$ m. If the latter, which $d$?

Lines 19-30 of this section explain the Monteith alternative, but in a rather confusing way. For example, when is $z_0$ ever zero, as stated on line 23, or why does $\partial_z R_a \to R_a$ for finite $z$? (I know what they intend to say, but it isn't at all clear.) In any case, here the authors end up with a stability-independent equation for $R_a$, without mentioning or discussing that fact.

This very shallow layer is also very problematic for deposition calculations in general, since the model cell seems to be run here with horizontal dimensions of $2.25 \times 2.25\,°$, or about 250 km $\times$ 250 km near the equator, but a vertical mid-level (CTM3's $z_{ref}$) of just 8 m. Now, profiles of wind and depositing gases are very sensitive to the underlying surface, and should be very different for forests or lakes for example. Any wind-speed or friction velocity calculated from a model of such large horizontal resolution will necessarily give values at 8 m which reflect the whole grid. Deposition rates for a specific land-cover will vary enormously depending on what else is in the grid-square. (Although not strictly comparable, we showed in Schwede et al. (2018) that differences between the grid-average and forest specific deposition rates of N-compounds could be as much as a factor of two and up to more than a factor of five in extreme cases. These differences were largely dependent on how much forest occupied each grid cell.)

**C) Why so much focus on sea areas?**

The text seems rather unbalanced with regard to the different land-covers. Sect. 2 uses 1/2 page on various $z_0$ corrections for oceans, but say nothing about the ecosystem where ozone is expected to deposit at high rates: forests, crops, and other terrestrial ecosystems. The supplementary has three Figs related to this oceanic deposition. Why?

**D) Use of term 'EMEP scheme'?**

Sect. 3 discusses the comparisons of Vg in terms of 'EMEP scheme' versus 'Wesely scheme', and sensitivity tests are named e.g. 'EMEP_offlight'. As noted above the scheme implemented in CTM3 is very different to that implemented in the EMEP model, so this is very misleading. Please rename your scheme to something else.

I am worried that readers might get the impression that it is the EMEP scheme which is being tested here, but it certainly is not. For those interested, the EMEP scheme itself, and its stomatal conductance formulation, have been discussed extensively in a series of papers, e.g. Emberson et al. (2000a,b, 2001), Klingberg et al. (2008), Simpson et al. (2001, 2003), Tuovinen et al. (2001, 2004, 2009).

**E) Reproduction of material from S2012**

As far as I can see, Table S1, S2 and S3 are taken directly from S2012, with no change to parameters. It is not usual to copy tables from the work of other authors in this way. Just refer to S2012 (and give Table number as help).

Many of the equations are from S2012, and many not. I would like the authors to make this very explicit, so that readers are not confused as to what comes from EMEP, and what has changed for CTM3.

**F) Other**

1. p1, line 22. Isn't $H_2O$ the most important greenhouse gas? (Say anthropogenic GHG perhaps?)

2. p2, line 3. The Wilson ref only concerns Europe, and its focus on the 95th percentile can hide trends found at higher percentiles (e.g. Simpson et al. 2014). A better ref would be Fleming et al. (2018) or Mills et al. (2018a). By the way, the most recent calculation on food security (using flux approaches) is now Mills et al. (2018b).

3. p2, line 11. What does *in situ* mean here? Ozone production can take place over days of transport.

4. p2, lines 25-35. This text about halogens is not really relevant to a dry deposition paper. Reactions with bromine can be important sinks, but are not usually counted as deposition.

5. p3, line 2-3. Why specify mid-June maximum for ozone. Monks (2000) might take issue with that, as would for example Sinha et al. (2015).

6. p3, line 4-5. There are plenty of ozone measurements made outside Europe. The authors appear to be unaware of the massive ozone collections made under the TOAR project (see e.g. Flemming, Mills refs below), or the high quality data available from GAW (inha 2015).

7. p3, line 16. One also has dry deposition to water, as this paper makes clear later on.

8. p3, line 20. One usually refers to dry deposition as something between a near-surface height (e.g. $z = 1m$, 10m, or 50m) and the surface, not from $z_0$. In fact, at $z = z_0$ one has $u_z = 0$, and hence the author's $R_a$ just below should be zero.

9. p4, line 20. I would remove the term textbook knowledge, since there are many different approaches to nearly all these equations. It is thus good that the equations as used in CTM3 are spelled out explicitly.

10. p5, line 5. I would add Emberson et al. (2000a) and Tuovinen et al. (2004) to the list of EMEP refs here, since this was the first publication of the methods that have more or less been used until today.

11. p7, notation. In S2012 and EMEP generally, we use upper-case $G$ and $R$ to refer to canopy-scale (bulk) variables, and lower-case for leaf-scale. Thus, in EMEP we would have $G_{sto} = \text{LAI } g_{sto}$. Here the authors seem to mix upper and lower case between their equations (13) and (14).

12. p7, Eqn (13). Is LAI one-sided, 2-sided, projected .... define.

13. p8, line 1. S2012 do not suggest using depths lower than 1m. We use SMI3 which is from 28-100 cm.

14. p8, line 2 - why did you choose to use the surface (0-7cm) soil moisture?

15. p8, line 18. This is wrong. Nothing in the EMEP model is used to 'mimic the time lag..'. We use the light function to modify stomatal conductance, as with the other $f$ factors.

16. p9, line 20, and Table 1. The consequence of Table 1 is that vegetation at 0.5° N will start growing at day 90, whereas those at 0.5° S will start on day 272. (By the way, in EMEP now we use monthly factors from the LPJ-GUESS model to derive phenology for non-European areas, because of such difficulties with tabulations.)

17. p9, line 18. what do you mean by "surface or 2m"? Surface might refer to skin or leaf temperature?

18. p9, Eqn (26). Say 1st and 2nd, not I. and II.

19. p9, Eqn (27). This equation is a modification of Erisman's original (1994) formulation, so explain that.

20. p10, line 12. Be explicit that this statement refers to S2012. The current EMEP model uses different heights for e.g. tropical vegetation.

21. p11, Sect 2.2. I also found this aerosol section confusing. Eqn (30) is from S2012, and so is the factor 0.008.SAI/10 used in Eqn (33), but here new $a_1$ coefficients are defined. Did the 'aerosol microphysic model' referred to also mix equations in this way? Is there any publication as to the reliability of this method?

22. p12, Fig.2. I didn't understand what is being done here. The Figure suggests that the EMEP scheme has one category for 'Forests, Med. scrub', whereas S2012 lists 4 types of forest, as well as Mediterranean scrub as a separate ecosystem. This figure also suggests that EMEP has savanna, which it doesn't, but do have many other categories (Table 3 of S2012 lists 16 main categories. The current EMEP model has 32.)

23. p12, line 13. Again, the current EMEP model is not eurocentric, and uses global phenology calculations.

24. p13, Sect 2.3.2. The initial lines (14-16) are hard to understand and only by reading further do I see what they mean by 'de-accumulated'. If working with IFS PPFD is so hard, why didn't the authors just calculate hourly (or minute-by-minute) PPFD using cloud-cover and zenith angles?

**References**

Emberson, L., Simpson, D., Tuovinen, J.-P., Ashmore, M., and Cambridge, H.: Towards a model of ozone deposition and stomatal uptake over Europe, EMEP MSC-W Note 6/2000, The Norwegian Meteorological Institute, Oslo, Norway, 2000a.

Emberson, L., Wieser, G., and Ashmore, M.: Modelling of stomatal conductance and ozone flux of Norway spruce: comparison with field data, Environ. Poll., 109, 393–402, 2000b.

Emberson, L., Ashmore, M., Simpson, D., Tuovinen, J.-P., and Cambridge, H.: Modelling and mapping ozone deposition in Europe, Water, Air and Soil Pollution, 130, 577–582, 2001.

Erisman, J. W., Van Pul, A., and Wyers, P.: Parametrization of surface resistance for the quantification of atmospheric deposition of acidifying pollutants and ozone, Atmos. Environ., 28, 2595–2607, 1994.

Fleming, Z. L., Doherty, R. M., von Schneidemesser, E., Malley, C. S., Cooper, O. R., Pinto, J. P., Colette, A., Xu, X., Simpson, D., Schultz, M. G., Lefohn, A. S., Hamad, S., Moolla, R., Solberg, S., and Feng, Z.: Tropospheric Ozone Assessment Report: Present-day ozone distribution and trends relevant to human health, Elem Sci Anth., 6(1):12, https://doi.org/doi.org/10.1525/elementa.273, 2018.

Garratt, J.: The atmospheric boundary layer, Cambridge University Press, Cambridge, England, 1992.

inha: The Global Atmosphere Watch reactive gases measurement network, Elementa-Science of the Anthropocene, 3, 000 067–Article, https://doi.org/10.12952/journal.elementa.000067, 2015.

Klingberg, J., Danielsson, H., Simpson, D., and Pleijel, H.: Comparison of modelled and measured ozone concentrations and meteorology for a site in south-west Sweden: Implications for ozone uptake calculations, Environ. Poll., 115, 99–111, 2008.

Mills, G., Pleijel, H., Malley, C., Sinha, B., Cooper, O. R., Schultz, M. G., Neufeld, H. S., Simpson, D., Sharps, K., Feng, Z., Gerosa, G., Harmens, H., Kobayashi, K., Saxena, P., Paoletti, E., Sinha, V., and Xu, X.: Tropospheric Ozone Assessment Report: Present-day tropospheric ozone distribution and trends relevant to vegetation., Elem. Sci. Anth., 6, https://doi.org/10.1525/elementa.302, `https://doi.org/10.1525/elementa.302`, 2018a.

Mills, G., Sharps, K., Simpson, D., Pleijel, H., Broberg, M., Uddling, J., Jaramillo, F., Davies, William, J., Dentener, F., Berg, M., Agrawal, M., Agrawal, S., Ainsworth, E. A., Büker, P., Emberson, L., Feng, Z., Harmens, H., Hayes, F., Kobayashi, K., Paoletti, E., and Dingenen, R.: Ozone pollution will compromise efforts to increase global wheat production, Global Change Biol., 24, 3560–3574, https://doi.org/10.1111/gcb.14157, `https://onlinelibrary.wiley.com/doi/abs/10.1111/gcb.14157`, 2018b.

Monks, P.: A review of the observations and origins of the spring ozone maximum, Atmos. Environ., 34, 3545–3561, https://doi.org/10.1016/S1352-2310(00)00129-1, 2000.

Schwede, D. B., Simpson, D., Tan, J., Fu, J. S., Dentener, F., Du, E., and deVries, W.: Spatial variation of modelled total, dry and wet nitrogen deposition to forests at global scale, Environ. Poll., 243, 1287 – 1301, https://doi.org/https://doi.org/10.1016/j.envpol.2018.09.084, `http://www.sciencedirect.com/science/article/pii/S0269749118327386`, 2018.

Simpson, D., Tuovinen, J.-P., Emberson, L., and Ashmore, M.: Characteristics of an ozone deposition module, Water, Air and Soil Pollution: Focus, 1, 253–262, 2001.

Simpson, D., Tuovinen, J.-P., Emberson, L., and Ashmore, M.: Characteristics of an ozone deposition module II: sensitivity analysis, Water, Air and Soil Pollution, 143, 123–137, 2003.

Simpson, D., Benedictow, A., Berge, H., Bergström, R., Emberson, L. D., Fagerli, H., Flechard, C. R., Hayman, G. D., Gauss, M., Jonson, J. E., Jenkin, M. E., Nyíri, A., Richter, C., Semeena, V. S., Tsyro, S., Tuovinen, J.-P.,

Valdebenito, A., and Wind, P.: The EMEP MSC-W chemical transport model – technical description, Atmos. Chem. Physics, 12, 7825–7865, https://doi.org/10.5194/acp-12-7825-2012, `http://www.atmos-chem-phys.net/12/7825/2012/acp-12-7825-2012.html`, 2012.

Simpson, D., Arneth, A., Mills, G., Solberg, S., and Uddling, J.: Ozone - the persistent menace; interactions with the N cycle and climate change, Current Op. Environ. Sust., 9-10, 9–19, https://doi.org/http://dx.doi.org/10.1016/j.cosust.2014.07.008, sI: System dynamics and sustainability, 2014.

Sinha, B., Sangwan, K. S., Maurya, Y., Kumar, V., Sarkar, C., Chandra, B. P., and Sinha, V.: Assessment of crop yield losses in Punjab and Haryana using 2 years of continuous in situ ozone measurements, Atmos. Chem. Physics, 15, 9555–9576, https://doi.org/10.5194/acp-15-9555-2015, 2015.

Tuovinen, J.-P., Simpson, D., Mikkelsen, T., Emberson, L. D., Ashmore, M. R., Aurela, M., Cambridge, H. M., Hovmand, M. F., Jensen, N. O., Laurila, T., Pilegaard, K., and Ro-Poulsen, H.: Comparisons of measured and modelled ozone deposition to forests in Northern Europe, Water, Air and Soil Pollution: Focus, 1, 263–274, https://doi.org/10.1023/A:1013131927678, `https://doi.org/10.1023/A:1013131927678`, 2001.

Tuovinen, J.-P., Ashmore, M., Emberson, L., and Simpson, D.: Testing and improving the EMEP ozone deposition module, Atmos. Environ., 38, 2373–2385, 2004.

Tuovinen, J.-P., Emberson, L., and Simpson, D.: Modelling ozone fluxes to forests for risk assessment: status and prospects, Annals of Forest Science, 66, 401, `dx.doi.org/10.1051/forest/2009024`, 2009.

---

## Referee Comment (RC3) · Anonymous Referee #1 · 1 May 2019

The authors state in their response that "In fact, we originally used Eq. (2) with L in the denominator for the second term. The sign error was likely the reason why the Monteith method was chosen. Certainly, an update shall be considered in the future, but it is not feasible to redo all simulations now."

The fact of the matter is that there have been some fundamental inconsistencies in the formulation of this equation, and the authors' logic of "it is not feasible to redo all simulations now" is not tenable. Ideally, the simulation would need to be done again. At the very least, the authors need to perform some test runs which demonstrate whether or not the inconsistencies in Eq. (2) make any significant difference to the results.

The authors have also not clarified how the value of the zero-plane displacement height d is selected. They simply state that d is constant (typically 0.7m). Looking up any

atmospheric boundary-layer text book (e.g. Garratt, 1992), d is approximately 0.7 times the canopy height (or the height of the roughness element). Please check this for consistency too.
* * *

---

## Author Response (AR1)

**Final response**

We thank both referees for their thorough review of our paper and the evaluation of the formulation of the dry deposition scheme. In consideration of all comments, it became necessary to revise a major part of the dry deposition implementation in our model. Particularly, we

- have addressed and corrected the equations in

  - the implementation of the mosaic approach,
  - the computation of $R_a$.

- have fixed the equations with respect to comprehensibility (e.g., indexing),

- have included a diagnostic output of dry deposition velocities,

- now refer to our new dry deposition scheme as *mOSaic scheme*,

- have refined the definition of our model experiments to make them more consistent, especially

  - using the soil moisture at a more appropriate level (SWVL3) and
  - CEDS emissions in accordance to the meteorological year.

  This also includes replacement or removal of some previous experiments

  - *EMEP_SWVL4 → mOSaic_SWVL1*,
  - ,
  - , and
  - *EMEP_ppgs_2005 → mOSaic_emis2014*.

  We have added two experiments in which we explicitly change the prescribed ozone surface resistance $R^{O_3}$:

  - *mOSaic_desert*,
  - *mOSaic_hough*.

- have elaborated on the evaluation and discussion of the results and

- included a section in which we compare our results to the MACC-reanalysis.

In the following, we account for all referee comments in detail and include the track-change-version of the manuscript. **Re-**
25 **mark:** We have not updated the responses that have been already published in gmdd (RC1, RC3). Hence, the referenced values therein differ from the final manuscript.

**Authors' response**

To gmd-2019-21 Anonymous Referee #1 (02 Apr 2019): We thank the referee for his/her useful comments and we will take them into account in the revised version of this paper. In the following, we will respond to the questions in detail.

- Abstract L15–17: *"While high sensitivity to changes in dry deposition to vegetation is found in the tropics, the largest impact on global scales is associated to changes in dry deposition to the ocean and deserts."* The authors do not provide details in the paper as to what has changed in the updated scheme for such an impact. We elaborate on the source of the *"largest impact on global scales"* in Section 3.2.2 (P18 L4–11). But we have to admit that it might not be clear to which resistance term the changes in the prescribed dry deposition velocities apply.

    - Is it the surface resistance ($R_c$) value? Or the other two resistances ($R_a$ and $R_b$)? What are the typical values? It would be indeed interesting to look at the resistance terms separately. However, they are not available in our output files. Technically, it would be possible to force the model output but such would involve redoing the experiments and run for at least a couple of (model) weeks. In our formulation, the surface resistance $R_c$ includes both stomatal and non-stomatal conductance. In case, of non-vegetated surfaces, such as *desert*, *ocean*, and *snow/ice*, the non-stomatal conductance, $G_{ns}^{O_3}$, is dominant. For water, $R_b = 10\,\mathrm{s\,m^{-1}}$ in most cases which is 2 orders of magnitude smaller than $R_c$. Thus, $R_{gs}^{O_3} \approx R_c^{O_3} \propto (v_{DD}^{O_3})^{-1}$ (for $R_c$ values see Table S2 in the manuscript Supplement S.3).

    - What value of $R_c$ for water has been used and how it compares with the value used in the Wesely scheme? We have given the prescribed dry deposition velocities in Section 3.2.2:
        - For the *Wesely scheme* $v_{water}^{O_3} = 0.07\,\mathrm{cm\,s^{-1}}$ and
        - for the *EMEP scheme* $v_{water}^{O_3} = 0.05\,\mathrm{cm\,s^{-1}}$.

    Surface resistances for water are thus $R_c \approx 1429\,\mathrm{s\,m^{-1}}$ (*Wesely scheme*) and $R_c = 2000\,\mathrm{s\,m^{-1}}$ (*EMEP scheme*), respectively.

- P22 Table 3:

    - Why the deposition values for ocean + ice + land do to add up to the total values reported, for all simulations? Yes, that is, for we exclusively selected gridboxes associated to more then 98% to the three surface types (ocean, ice, and land), while *total* comprises all gridboxes. We admit that this is not clear. We shall repeat the computation of the values using the proper weighting and adapt the numbers (Table 5). We may also add that *ice* in this analysis only refers to regions at high latitudes that are permanently covered by ice and snow and hence does not take sea ice and mountain glaciers into account. In the Oslo CTM3, as written elsewhere in the manuscript, we compute dry deposition on ice and snow based on meteorological data. Therefore, the given values in Table 3 comprise ozone dry deposition on sea ice in the case of *ocean* and on snow covered land during winter in case of *land*.

    - The new land-based deposition values are much lower than what has been reported in previous studies (e.g. Hardacre et al., 2015) and the authors largely attribute this to the changes in the updated scheme for the desert surface type. However, the paper does not provide any observational support to back this up.

        - Are there any relevant deposition measurements (velocity or flux) that can be used for this purpose? Güsten et al. (1996) conducted measurements of ozone concentrations and fluxes onto the Sahara desert and deduced dry deposition velocities. We have not found any recent paper conducting field experiments in desert regions. We include a more thorough discussion of our results with respect to the observed fluxes by Güsten et al. (1996) and the references therein the revision of our manuscript.

        - At least, some comparison with ozone measurements (or even $O_3$ reanalyses) should be provided for this surface type (and perhaps others) to see if the model is heading in the right direction with the updated deposition scheme. Thank you for the advise. We will look into this.

    - It will also be useful to report the global ozone burden from the various simulations. Since Table 3 is already at maximum width with respect to the the page width, we shall show the global tropospheric ozone burden in a

**Table 1.** Total ozone dry deposition for the respective model experiment in $\mathrm{Tg\,a^{-1}}$. The global ozone dry deposition has been weighted by ocean, ice and, land fraction in each gridbox, respectively. *Ice* herein refers to regions at high latitudes that are permanently covered by ice and snow.

| Experiment | Ocean | | | Ice | | | Land | | |
|---|---|---|---|---|---|---|---|---|---|
| | NH | SH $(\mathrm{Tg\,a^{-1}})$ | Global | NH | SH $(\mathrm{Tg\,a^{-1}})$ | Global | NH | SH $(\mathrm{Tg\,a^{-1}})$ | Global |
| Wesely_type | 159.7 | 147.8 | 307.5 | 3.6 | 6.3 | 9.9 | 446.6 | 193.7 | 640.3 |
| EMEP_full | 107.6 | 105.2 | 212.8 | 2.5 | 5.5 | 8.0 | 296.6 | 135.0 | 431.6 |
| EMEP_offLight | 110.0 | 105.8 | 215.8 | 2.5 | 5.4 | 7.9 | 337.1 | 156.0 | 493.0 |
| EMEP_offPhen | 108.0 | 105.3 | 213.2 | 2.5 | 5.5 | 8.0 | 301.6 | 139.1 | 440.7 |
| EMEP_SWVL4 | 109.2 | 107.0 | 216.2 | 2.6 | 5.5 | 8.1 | 303.6 | 138.2 | 441.8 |
| EMEP_ppgs | 107.6 | 105.2 | 212.7 | 2.5 | 5.5 | 8.0 | 299.5 | 135.3 | 434.9 |
| EMEP_ppgssh | 108.3 | 105.4 | 213.6 | 2.5 | 5.5 | 8.0 | 306.8 | 142.8 | 449.6 |
| EMEP_ppgssh_ice | 107.4 | 105.3 | 212.7 | 1.2 | 1.6 | 2.8 | 303.0 | 144.1 | 447.2 |
| EMEP_ppgs_2005 | 106.9 | 103.5 | 210.4 | 2.6 | 5.4 | 8.0 | 297.9 | 131.8 | 429.7 |

separate table (see the following Table 6). If we compare our results with Stevenson et al. (2006) ($344 \pm 39\,\mathrm{Tg}$) and the number given in IPPC AR5 (2013) ($337 \pm 23\,\mathrm{Tg}$), we find that the ozone burden in the Oslo CTM3 is higher then the model average from the start (*Wesely scheme*). The implementation of the *EMEP scheme* increases the tropospheric burden by roughly 8 % (compare *Wesely type* and EMEP_full).

**Table 2.** Annual mean tropospheric ozone burden for all experiments and $1\sigma$ standard deviation.

| Experiment | Trop. $O_3$ (Tg) | | |
|---|---|---|---|
| Wesely type | 361 | $\pm$ | 21 |
| EMEP_full | 392 | $\pm$ | 28 |
| EMEP_offLight | 388 | $\pm$ | 26 |
| EMEP_offPhen | 392 | $\pm$ | 27 |
| EMEP_SWVL4 | 402 | $\pm$ | 31 |
| EMEP_ppgs | 392 | $\pm$ | 28 |
| EMEP_ppgssh | 391 | $\pm$ | 27 |
| EMEP_ppgssh_ice | 403 | $\pm$ | 31 |
| EMEP_ppgs_2005 | 386 | $\pm$ | 26 |

5   – Section 2.1.1, Eq. (2): The statement *"For certain values of z, z0, and L, this may result in nonphysical (negative) values for Ra."* I do not comprehend as to why this would occur since this equation is simply based on the well-used Monin-Obukhov similarity theory (MOST) for the surface layer. This occurrence would also imply negative wind speeds. Actually Eq. (2) is incorrect: the term $\psi_m((z-d)/z_0)$ should be $\psi_m((z-d)/L)$, and the sign of the third term on the right-hand side should be positive (not negative). Given that $(z-d) > z_0$ (assuming the model is formulated correctly),

10   Eq. (2) should always yield positive values.
Eqs. (3–5): I am not sure why Monteith (1973) needs to be invoked here. Given that the term in the square brackets on right hand side of Eq. (2) is equal to $k \cdot u(z)/u^*$ as per MOST, substituting this into Eq. (2) results in Eq. (5). Define $z$, $z_0$ in Eq. (2). The parameter $d$ is the so-called displacement height, and is not a constant (depends on the surface type). The reviewer is indeed correct that this equation is wrong. In fact, we originally used Eq. (2) with $L$ in the denominator for

15   the second term. The sign error was likely the reason why the *Monteith method* was chosen. Certainly, an update shall be considered in the future, but it is not feasible to redo all simulations now. In the revised version of the manuscript, we change the text from *"In Simpson et al. (2003,2012) it is described as [...] fall back to the [...]"* to *"For technical reasons, we have used the [...]"*.

– P2 L25-33: The first reference to the Oslo CTM3 in the body of the paper is made here as *"...we have not implemented any parameterization of these processes in the Oslo CTM3 as of now."*

  – Some brief introductory text is required here (or better at the start of the paragraph) to introduce the model properly. The Oslo CTM3 is properly introduced in Section 2. Hence, we move the sentence in L30-32 about *polar boundary layer ozone depletion* to Section 2: *"Although the ozone depleting events in the polar boundary layer (Section 1) are important to understand surface ozone abundance in Arctic regions in spring-time, no parameterization of these processes is implemented in the Oslo CTM3 as of now."*

  – Also, the text between lines 25 – 33 on what is not considered in the model is too detailed to be here, so shorten and move it to Section 2. Given that the influence of VSLS on tropospheric ozone is indirect (through depletion of ozone in the upper troposphere – lower stratosphere and subsequent STE), this reference (L32–33) rather belongs to the discussion in Section 4 and will be moved: *"In particular, the STE depends on the stratospheric ozone abundance which is, e.g. affected by very short-lived ozone depleting substances (VSLS) (Warwick et al., 2006; Ziska et al., 2013; Hossaini et al., 2016; Falk et al., 2017) and is not taken into account in the Oslo CTM3."*

– P3 L19/L28 and P21 L34: There is a newer ozone dry deposition study by Luhar et al. (2018, ACP, 18, 4329-4348) which, using global ozone reanalyses and a more realistic process-based oceanic deposition scheme, estimates the total global deposition at $722.8 \pm 87.3 \, \mathrm{TgO_3yr^{-1}}$, which includes an oceanic component of $98.4 \pm 30.0 \, \mathrm{TgO_3yr^{-1}}$. These figures should be cited for comparison. Thank you for pointing this out. We were not aware of this study and will compare our results and refer to it at the given places and within our discussion.

  – *"A newer study by Luhar et al. (2018), however, indicates much lower amounts $(722.8 \pm 87.3 \, \mathrm{Tg\,a^{-1}})$."*

  – *"Based on the global atmospheric composition reanalysis performed in the ECWMF project Monitoring Atmospheric Composition and Climate (MACC) and a more realistic process-based oceanic deposition scheme, Luhar et al. (2018) found that the ozone dry deposition to oceans amounts to $98.4 \pm 30.0 \, \mathrm{Tg\,a^{-1}}$."*

  – *"But also the results of Luhar et al. (2017, 2018) yield a $(19-27)\,\%$ lower ozone dry deposition than the models participating in the model intercomparison, with deposition to ocean ranging between $(12-21)\,\%$ of the total annual ozone dry deposition."*

– P11 Section 2.2: Since the present paper is about ozone dry deposition, this section seems like a distraction and hence should be omitted. The referee is right in his/her assessment. We therefore omit this section in the revised version of our manuscript.

– P14 L15-18: Anthropogenic, biomass burning, and biogenic emissions are included in the model. How are other emissions such as soil $NO_x$, wetland methane, and oceanic methane and CO specified? Emissions from soil and wetlands are computed by MEGAN. Resultant $NO_x$ emissions are upscaled to match Global Emissions InitiAtive (GEIA) inventory. For oceanic emissions of CO, we use predefined global fields (POET, available through ACCENT/GEIA, http://accent.aero.jussieu.fr/database_table_inventories.php). $CH_4$ is taken from surface data from the EU project Hydrogen, Methane and Nitrous oxide: Trend variability, budgets and interactions with the biosphere (HYMN; EU GOCE 037048) for the year 2003 and scaled to oceanic amounts of $CH_4$ from NASA (https://www.esrl.noaa.gov/gmd/ccgg/trends_ch4/) given for the years 2000–2004. We shall include this information in the revised manuscript.

– P15 L4: The statement *"Accidentally, we have used emissions for the year 2014 instead of 2005."* It is not clear what the consequences on the results are of this? In the introduction to Section 3.1, we wrote *"For all model integrations, the meteorological reference year is 2005. This choice only affects the comparison with data and multi-model studies that either perform analysis on decadal averages or differing years."*. We will clarify the sentence with respect to probable consequences.

– Section 3.2.1:

- Section 3.2.1: In the Fig 5 discussion, although snow and ice is discussed, there is no discussion on the oceanic differences between the present study and Hardacre et al. (2015). This is particularly important for the Southern latitudes. We agree that ozone dry deposition to oceans is of high importance for the southern hemisphere, last but not least in the zonal band $(50-70)°S$. In fact, we have mainly discussed out results with respect to seasonal cycles of dry deposition velocities onto ocean found in Hardacre et al. (2015) (Fig. 7; P21 L1–10: *"Similarly, the dry deposition velocities over water differ. [...]"*). We will include a discussion based on Fig. 5 in the revision of the manuscript.

- The Hardacre et al. (2015) simulations were for the year 2001, whereas the present study is mostly for the year 2015 emissions (see Table 1) driven by the year 2005 meteorology. In addition, the observational averages used in Fig. 8 are based on multi-year data. The authors should discuss the implications of these differences about different years on the deposition results presented (e.g. uncertainty). We regard this remark as a follow-up of the question raised regarding P15 L4. We will elaborate on the discussion regarding the implications based on our model results (EMEP_ppgs vs EMEP_ppgs_2005) in the revision of the manuscript. Though, the major consequence of this is that, for the majority of our model experiments, one can neither directly compare to observations for the years 2005 and 2014 nor to other the model results. Emission inventories may always be lacking in certain chemical species and the non-linearities in ozone formation and destruction make ozone concentrations sensitive to both emissions of precursors and meteorological conditions. Surface ozone observations, in fact, show a strong variability in ozone dry deposition and ozone concentrations at the sites. But studying these in detail may be well beyond the scope of this manuscript.

– P24 L3–4: *"The annual amount of ozone dry deposition decreases by up to 100% changing from the old dry deposition scheme to the new one."* Table 3 does not support this, but this may be true for some surface types. So please qualify the statement. We have indeed not specified our statement in the mentioned sentence, while we had done so elsewhere in the manuscript (P15 L12). We complete our statement in the revised version of the manuscript: *"[...] ozone dry deposition decreases by up to $100\%$ over all major desert areas [...]. At the same time, it increases over tropical forest.*

– P24 L15: *"Most of the decrease in ozone dry deposition in the Oslo CTM3 can be attributed to changes in dry deposition velocities over the ocean and deserts."* What are the dominant factors in these changes? For example, is it mostly the surface resistance ($R_c$) term? For the ocean, it is likely to be $R_c$. For deserts, maybe $R_b$? Is it possible to quantify these differences in the resistance terms? We have already answered the question with respect to ocean (see first bullet point). In summary, since $R_b$ is quite small in most of the cases, the dominant factor for the ozone dry deposition onto ocean is the surface resistance $R_c$ which is tabulated in Table S2. Regarding ozone dry deposition onto deserts, we use Eqs. (7–8) to deduce

$$R_b^i = \frac{2}{\kappa u_*} \cdot \left( \frac{D_{H_2O}}{D_i} \cdot \frac{Sc_{H_2O}}{Pr} \right)^{\frac{2}{3}},\tag{1}$$

with $Pr = 0.72$, $\kappa = 0.4$, $Sc_{H_2O} = 0.6$, $D_{H_2O}/D_i = 1.6$. We estimate $u_*$ from Eq. (16.67) in Seinfeld and Pandis (2006)

$$u_*(h) = \frac{\kappa \cdot \overline{u_x}(h)}{\ln(h/z_0)},\tag{2}$$

with $h = 8\,\mathrm{m}$, $z_0^{\mathrm{desert}} \approx 10^{-3}\,\mathrm{m}$, and wind speeds not exceeding a gentle breeze ($1.8\,\mathrm{km\,h^{-1}} \le \overline{u_x}(h) \le 28\,\mathrm{km\,h^{-1}}$), we find $272\,\mathrm{s\,m^{-1}} \ge R_b \ge 17\,\mathrm{s\,m^{-1}}$. This is $1-2$ orders of magnitude smaller than $R_c = 2000\,\mathrm{s\,m^{-1}}$ and thus not negligible for low wind speeds. In summary, $R_c$ is dominant in our formulation of dry deposition of ozone to deserts (unless we have calm wind conditions).

– P24 L24: *"2-layer gas exchange with ocean waters (Luhar et al., 2017)."* As mentioned earlier, Luhar et al. (2018) has derived a more realistic process-based deposition scheme for the ocean, but the results for deposition velocity do not seem to be too different from those in Luhar et al. (2017). We acknowledge Luhar et al. (2018) an update the sentence: *"[...] 2-layer gas exchange with ocean waters (Luhar et al., 2017, 2018)."*

- P25 L11–12: The comment *"This is most likely reflecting the ongoing industrialization process of countries in the southern hemisphere and the commitment and implementation of air quality regulations of industrialized nations in the northern hemisphere"* is quite speculative and may be omitted. We follow the kind advise of the referee and remove the sentence in the revised version of the manuscript.

- Eq. (13) cf. Eq. (14): $g_{STO}$ or $G_{sto}$ – use consistency with notation. Thanks for pointing this out. We will change this in the revised version of the manuscript.

- The first half of the abstract, the text before *"In this paper...,"* is introductory material and can be deleted. This is indeed the case and we will remove it in the revised version.

- Abstract L15–16: it is better to say "...leading to an increase in surface ozone of up to 100% in some regions." We follow the advice of the referee and change the sentence accordingly.

- P22 L7: *"At about 4 of 6 sites."* About? Not sure? Thanks for pointing out the misplaced *"about"* in this sentence. We are certain regarding that number.

**Authors' response**

To gmd-2019-21 referee#2 David Simpson (10 Apr 2019): We apologize for the confusion the misuse of the terms *EMEP* and *EMEP scheme* may have caused. We evaluated the major concerns raised in the review and found them severe enough that we have to revise our model and repeat the model experiments to address all concerns. Nevertheless, we will respond to all of the questions and concerns in more detail in the following.

**Major points**

– Mosaic formulation:

– CTM3 claims to implement a mosaic approach, but instead of calculating deposition rates over each land-cover, and then aggregating using Eqn.3, CTM3 seems to perform the following steps:

$$R_a = \frac{u_z}{u_*^2} \tag{3}$$

$$\overline{Gs_g^i(z)} = \Sigma_{k=1}^N f_k \times Gs_{g,k}^i(z) \tag{4}$$

$$\overline{Gns_g^i(z)} = \Sigma_{k=1}^N f_k \times Gns_{g,k}^i(z) \tag{5}$$

As far as I can tell from the text, $R_a$ is calculated once, with the same value of $u_*$ for all land-covers. The CTM3 $R_b$ calculation seems to also use the same $u_*$ over different land-covers, except over sea where a more sophisticated scheme is used. Equations 2–3 above are equivalent to Eqns (22) and (26) from the manuscript. Now I am puzzled however as to how all this is put together. Do they use:

$$G_c = \mathrm{LAI} \cdot \overline{Gs_g^i(z)} + \overline{Gns_g^i(z)} \tag{6}$$

Thank you very much for your detailed account of the *mosaic approach*. It is correct that we use Eq. (4) and Eq. (5). As pointed out, we have not properly indicated the weighted mean in Eq. (22) and Eq. (29) in the manuscript. Additionally, there is also a typo in Eq. (13) and Eq. (22) ($G_{\mathrm{sto}}$ should have been $g_{\mathrm{sto}}$). Eq. (22) should have read as follows:

$$\overline{g_{\mathrm{sto}}^{\mathrm{m}}} = \sum_{n=0}^N f_{\mathrm{L},n} \cdot g_{\mathrm{sto},n}^{\mathrm{m}} \tag{7}$$

All is put together in Eq. (13):

$$G_c = \mathrm{LAI} \cdot \overline{g_{\mathrm{sto}}^{\mathrm{m}}} + \overline{G_{\mathrm{ns}}} \tag{8}$$

We acknowledge that Eq. (13) should not proceed the other equations and that this might have caused additional confusion.

– In any case, I think this approach has serious problems. Why average first for $G_s$ and then for $G_{ns}$, when it is the fluxes ($F_k$, or $V_{g,k}^i(z_{\mathrm{ref}}) \times \chi_{\mathrm{avg}}^i(z_{\mathrm{ref}})$) which need to be averaged? I also do not understand why they would use the same $u_*$ and $R_a$ for all land-covers. I think the authors need to make a case for their approach, or change it. We have discussed the concerns raised and found that we have indeed made a mistake in our implementation of the *mosaic approach* which forces us to revise our model and repeat the model experiments. We update the equations in the revised manuscript accordingly.

– Calculation of $R_a$:

– $R_a$ in CTM3 appears to be calculated just once, and from a height of 8 m. This means that there is no consistency between the $R_a$ term and the underlying surface, which is clearly wrong. The similarity equation for $R_a$ given in their Eqn (2) is very standard and has been used for decades (Garratt 1992). As pointed out by Ref 1, the equation as given has errors. Eq. (2) in our manuscript is indeed erroneous. The correct equation is (e.g., Erisman, 1994):

$$R_a = \frac{1}{\kappa u_*} \left[ \ln\left(\frac{z-d}{z_0}\right) - \Psi_m\left(\frac{z-d}{L}\right) + \Psi_m\left(\frac{z_0}{L}\right) \right]. \tag{9}$$

– The correct equation will not give negative values unless presented with wrong inputs, and I suspect that that is what has happened. It is actually difficult to tell what was tested from the manuscript though, since they state simply that $d$ is 'typically 0.7 m'. Did they use $d$ properly, consistent with the underlying land-cover and its $z_0$? Did they assume that their 8 m meteorology was at a physical height of 8 m, or at $d+8$ m? If the latter, which $d$? At that point, it is indeed neither clear from the manuscript nor from our internal documentation. The assumption that Eq. (2) (in the manuscript) would become negative, was most likely based on a wrong form of the integrated stability function $\Psi_m$.

Given $z = z_{\text{ref}} = 8$ m and $d = 0.7 \cdot 1$ m, it is right that neither the correct Eq. (9) nor the erroneous equation will result in negative $R_a$. The statement "$d$ is 'typically 0.7 m'" is incorrect, indeed. We correct: "[...] $d_k = 0.78 \cdot h_k(\text{lat})$, $z_0^k = 0.07 \cdot h_k(\text{lat})$ *for forests*, $d_k = 0.7 \cdot h_k(\text{lat})$, $z_0^k = 0.1 \cdot h_k(\text{lat})$ *for vegetation other than forests [...]*". Side note: The average height of the lowermost model level is actually 20 m (10 m for mid-level).

– Lines 19–30 of this section explain the Monteith alternative, but in a rather confusing way. For example, when is $z_0$ ever zero, as stated on line 23, or why does $\partial_z R_a \rightarrow R_a$ for finite $z$? (I know what they intend to say, but it isn't at all clear.) The referee is right. The text and derivation of the Monteith relation is confusing and also erroneous. In a revised version of the dry deposition scheme, we do no longer use these formulations, hence they are removed from the revised manuscript. Nevertheless, the correct form should have read as follows:

$$\Phi_{Q_{\text{sensible}}} = \rho_{\text{air}} c_P \cdot \frac{\Delta T}{R_{a,H}}. \tag{10}$$

$$\Phi_{Q_{\text{sensible}}} = \rho_{\text{air}} c_P \cdot \frac{\partial T}{\partial u} \cdot u_*^2. \tag{11}$$

For finite $z$, $\partial T \rightarrow \Delta T$.

– In any case, here the authors end up with a stability-independent equation for $R_a$, without mentioning or discussing that fact. In the revision of the model, we are going to implement the stability dependent $R_a$ following Eq. (9) and update the manuscript accordingly.

– This very shallow layer is also very problematic for deposition calculations in general, since the model cell seems to be run here with horizontal dimensions of $2.25 \times 2.25°$, or about $250 \times 250$ km near the equator, but a vertical mid-level (CTM3's $z_{\text{ref}}$) of just 8 m. Now, profiles of wind and depositing gases are very sensitive to the underlying surface, and should be very different for forests or lakes for example. Any wind-speed or friction velocity calculated from a model of such large horizontal resolution will necessarily give values at 8 m which reflect the whole grid. Deposition rates for a specific land-cover will vary enormously depending on what else is in the grid-square. (Although not strictly comparable, we showed in Schwede et al. (2018) that differences between the grid-average and forest specific deposition rates of N-compounds could be as much as a factor of two and up to more than a factor of five in extreme cases. These differences were largely dependent on how much forest occupied each grid cell.) Thank you for pointing this out. Indeed, an evaluation of winds at 8 m does not make much sense, given that forests in the tropics reach heights up to 40 m. In S2012, $R_a$ is actually evaluated "at around 45 m" which is similar to the center of our second lowest model level. We correct for this in the revised model version.

- Why so much focus on sea areas?: The text seems rather unbalanced with regard to the different land-covers. Sect. 2 uses 1/2 page on various $z_0$ corrections for oceans, but say nothing about the ecosystem where ozone is expected to deposit at high rates: forests, crops, and other terrestrial ecosystems. The supplementary has three Figs related to this oceanic deposition. Why? The referee is right that ozone is predominately deposited to terrestrial ecosystems, which cover about 2/9 of the Earth's surface, while 2/3 is covered by oceans. Nevertheless, due to the ocean's vastness, relatively small changes in modeled dry deposition easily accumulate and influence the atmospheric ozone concentration as pointed out in Hardacre et al. (2015) based on Ganzeveld et al. (2009). Luhar et al. (2017, 2018) show that the current models may overestimate the oceanic ozone sink by a factor of three – rendering the ocean a even smaller sink for ozone.

  This said, Section 2 as a whole consists of about 9 pages, of which 1/2 page is dedicated to oceans ($R_b$ computation) and 7 pages are dedicated to vegetation especially stomatal and non-stomatal conductance ($R_c$ computation). Hence, we do not see a significant unbalance in favor of oceans herein.

  Regarding the three figures with respect to ocean in the supplementary material: The reason for these is that we found some "legacy code" in the model (linear fit to dynamic viscosity of air $\mu(T)$) after we had finished the implementation of the new dry deposition scheme, run the experiments, and done the analysis on these. We had to verify that this had no significant influence on the results.

- Use of the term 'EMEP scheme'?: Sect. 3 discusses the comparisons of $V_g$ in terms of 'EMEP scheme' versus 'Wesely scheme', and sensitivity tests are named e.g. 'EMEP_offlight'. As noted above the scheme implemented in CTM3 is very different to that implemented in the EMEP model, so this is very misleading. Please rename your scheme to something else. I am worried that readers might get the impression that it is the EMEP scheme which is being tested here, but it certainly is not. [...] We apologize for the misleading naming of the new dry deposition scheme in the Oslo CTM3 which is (partly) based on the formulations in the publication the referee refers to as *S2012*. By no means have we meant to offend any of the original authors of the EMEP/MSC-W model nor intended to misguide the readers. We will rename the revised scheme appropriately ($\rightarrow$*mOSaic scheme*).

- Reproduction of material from S2012:

  - As far as I can see, Table S1, S2 and S3 are taken directly from S2012, with no change to parameters. It is not usual to copy tables from the work of other authors in this way. Just refer to S2012 (and give Table number as help). We will follow this advice and refer to the tables in S2012 accordingly.

  - Many of the equations are from S2012, and many not. I would like the authors to make this very explicit, so that readers are not confused as to what comes from EMEP, and what has changed for CTM3. We are going to elaborate on this matter in a revised version and refer to our equations' sources more properly.

**Minor points**

- P1 L22. Isn't $H_2O$ the most important greenhouse gas? (Say anthropogenic GHG perhaps?) Thank you for pointing this out. We follow the advice and write: *"[...] ozone is a potent anthropogenic greenhouse gas [...]"*

- P2 L3. The Wilson ref only concerns Europe, and its focus on the 95th percentile can hide trends found at higher percentiles (e.g. Simpson et al. 2014). A better ref would be Fleming et al. (2018) or Mills et al. (2018a). By the way, the most recent calculation on food security (using flux approaches) is now Mills et al. (2018b).

  Thank you for pointing out these publications. We take them into consideration, when revising the manuscript.

- P2 L11. What does in situ mean here? Ozone production can take place over days of transport. *In situ* typically means *on site, locally*. In this context, we used it exactly in this meaning. We elaborate on this in a revised version of the paper: *Elevated ozone levels at a site may originate from both, the local production of ozone from its precursors, which are transported, and from advection of ozone itself. Long-range ozone transport occurs regularly and might be most important in regions that else lack precursors. Tropospheric ozone is produced in complex photochemical cycles involving precursor gases [...]*

– P2 L25–35. This text about halogens is not really relevant to a dry deposition paper. Reactions with bromine can be important sinks, but are not usually counted as deposition. The removal of ozone from the Arctic boundary layer in the presence of halogens is indeed not part of ozone dry deposition and we remove this reference in this context. Although these processes are not relevant with respect to ozone dry deposition, they matter regarding the observed ozone abundances in the Arctics. One of the proposed mechanisms to trigger bromine explosions involves, e.g., an "initialization" by ozone dry deposition.

– P3 L2–3. Why specify mid-June maximum for ozone. Monks (2000) might take issue with that, as would for example Sinha et al. (2015). The referee has got a point here. Of cause the maximum of ozone in the annual cycle depends on the location. We remove this specification.

– P3 L4–5. There are plenty of ozone measurements made outside Europe. The authors appear to be unaware of the massive ozone collections made under the TOAR project (see e.g. Flemming, Mills refs below), or the high quality data available from GAW (inha 2015). Thank you for reminding us of the data collected under the TOAR project. We knew about this data set but have to admit that we have not made much use of it, yet. We have, however, not stated that there are, in general, no sites outside of Europe, but that long-term observations (started before the 1950s) are only found in Europe. Maybe the term "long-term" is not clear enough. We add: *"From the observational side, the number of long-term observations (started before the 1950s) is limited and restricted to mainly European sides."*

– P3 L16. One also has dry deposition to water, as this paper makes clear later on. That is indeed true, but we didn't want to name all possible surfaces onto which dry deposition takes place. We, hence, shall write: *"Removal of any substance from the atmosphere which is not involving rain, e.g., through gravitational settling or by uptake by plants, soil, and water, is referred to as dry deposition.*

– P3 L20. One usually refers to dry deposition as something between a near- surface height (e.g. z = 1m, 10m, or 50m) and the surface, not from $z_0$. In fact, at $z = z_0$ one has $u_z = 0$, and hence the author's $R_a$ just below should be zero. Thank you for pointing out this inaccuracy. We, of cause, meant "near-surface" height or lowermost model level (which is indeed not at $z_0 = 0$). We change the text accordingly: *"[...] dry deposition is a product between near-surface ozone concentration $[O_3](z_0)$ (e.g. the lowermost model level) [...] "*

– P4 L20. I would remove the term textbook knowledge, since there are many different approaches to nearly all these equations. It is thus good that the equations as used in CTM3 are spelled out explicitly. We follow the advice and remove the sentence in which the term occurs and write: *"[...] we will give a detailed account of the new dry deposition scheme and the equations that we use."*

– P5 L5. I would add Emberson et al. (2000a) and Tuovinen et al. (2004) to the list of EMEP refs here, since this was the first publication of the methods that have more or less been used until today. We have added the references regarding the EMEP MSC-W model. *"[...] we mainly follow Simpson et al. (2012) in their description of dry deposition used in the European Monitoring and Evaluation Programme (EMEP) MSC-W model (see also, Emberson et al., 2000; Simpson et al., 2003; Tuovinen et al., 2004;), [...]"*

– P7, notation. In S2012 and EMEP generally, we use upper-case G and R to refer to canopy-scale (bulk) variables, and lower-case for leaf-scale. Thus, in EMEP we would have $G_{sto} = LAI g_{sto}$. Here the authors seem to mix upper and lower case between their equations (13) and (14). We have indeed mixed up the cases here.

– P7 Eqn (13). Is LAI one-sided, 2-sided, projected .... define. LAI is one-sided taken from ISLSCP2 FASIR. We add this to the sentence: *"[...] LAI is the one-sided leaf area index taken from ISLSCP2 FASIR [...]"*

– P8 L1. S2012 do not suggest using depths lower than 1 m. We use SMI3 which is from 28-100 cm. Thank you for the clarification. This may have been a misunderstanding on our side. Since we have to repeat our simulations, we will choose the appropriate SWVL from the start.

– P8 L2. Why did you choose to use the surface (0–7 cm) soil moisture? Initially we choose 0–7 cm due to the original scope of this work within our research project, in which we study ozone concentrations and uptake by plants in boreal regions where the soil columns are expected to be more shallow then, e.g., at mid-latitudes. We expect the stomatal conductance in these environments to be more sensitive to precipitation. Using the actual water availability to vegetation, would be better, but this is not feasible without a proper land surface model.

– P8 L18. This is wrong. Nothing in the EMEP model is used to 'mimic the time lag..'. We use the light function to modify stomatal conductance, as with the other $f$ factors. We change this inadequate formulation: *"[...] this integrated photon flux is used to modify the stomatal conductance in response to light."*

– P9 L20, and Table 1. The consequence of Table 1 is that vegetation at $0.5°$ N will start growing at day 90, whereas those at $0.5°$ S will start on day 272. (By the way, in EMEP now we use monthly factors from the LPJ-GUESS model to derive phenology for non-European areas, because of such difficulties with tabulations.) Thank you for the remark. We see this problem. Using an actual land model would solve this, but at that point it is not feasible to integrate such in the Oslo CTM3.

– P9 L18. What do you mean by "surface or 2 m"? Surface might refer to skin or leaf temperature? This may have been indeed unclear. It is the 2 m temperature. We change the text accordingly: *"[...] $\theta_{2m}$ is the 2 m temperature in °C."*

– P9 Eqn (26). Say 1st and 2nd, not I. and II. We follow the advise regarding Eq. (26) and also change Table 1 accordingly.

– P9 Eqn (27). This equation is a modification of Erisman's original (1994) formulation, so explain that. We follow the advise and write: *"The in-canopy resistance $R_{inc}$ (Erisman et al., 1994) is then modified with respect to each (vegetated) land type N [...]"* We have added the respective reference in the text.

– P10 L12. Be explicit that this statement refers to S2012. The current EMEP model uses different heights for e.g. tropical vegetation. We follow this advice and write: *"The vegetation height $h(N, lat)$ as described by Simpson et al. (2012) [...]"*

– P11 Sect 2.2. I also found this aerosol section confusing. Eqn (30) is from S2012, and so is the factor 0.008.SAI/10 used in Eqn (33), but here new $a_1$ coefficients are defined. Did the 'aerosol microphysic model' referred to also mix equations in this way? Is there any publication as to the reliability of this method? The $a_1$ coefficients were calculated from deposition velocities taken from the Oslo CTM2 (Oslo CTM2 list of publications, online). We meant to use this new scheme for aerosols in the Oslo CTM3, but as of now it has not been evaluated and is actually not used. Hence, we will remove this section from the paper and add: *"In addition to the gaseous species, Simpson at al. (2012) also modify aerosol deposition velocities, namely black carbon (BC) and organic carbon (OC), sulfuric aerosols ($SO_4$, MSA) and secondary organic aerosols (SOA), but we have not updated our model with respect to these.".*

– P12 Fig.2. I didn't understand what is being done here. The Figure suggests that the EMEP scheme has one category for 'Forests, Med. scrub', whereas S2012 lists 4 types of forest, as well as Mediterranean scrub as a separate ecosystem. This figure also suggests that EMEP has savanna, which it doesn't, but do have many other categories (Table 3 of S2012 lists 16 main categories. The current EMEP model has 32.) The caption may become clearer after we changed the name of our scheme, so that it can no longer be confused with the actual EMEP model. The purpose of the mapping is to project the more detailed categories for stomatal conductance to the ground-surface resistance categories (Table S19 in S2012 supplement). We have updated figure and text in accordance to the corrections made in the model code. It should be clearer now.

– P12 L13. Again, the current EMEP model is not eurocentric, and uses global phenology calculations. We follow the advice and drop the term "eurocentric" the text: *"Since the parameterization of SGS and EGS in Simpson et al. (2012) is not applicable [...]"*

– P13 Sect 2.3.2. The initial lines (14-16) are hard to understand and only by reading further do I see what they mean by 'de-accumulated'. If working with IFS PPFD is so hard, why didn't the authors just calculate hourly (or minute-by- minute)

PPFD using cloud-cover and zenith angles? We elaborate on the comprehensibility of this paragraph: *"From OpenIFS an accumulated surface PAR is available. It is integrated both, spectral (presumably $400 - 60\,\mathrm{nm}$) and temporally. For practical use in Eq. (18), we de-accumulate this field with respect to time and refer to the result as PPFD."* We have not used that ansatz, because we do not have the radiation fields in the relevant wavelengths available from OpenIFS output.

5 **Authors' response**

To gmd-2019-21 Anonymous Referee #1 (01 May 2019): We thank the referee for his comments. Taking all comments from both referees into consideration, we find it necessary to revise our model and redo the simulations.

- The authors state in their response that *"In fact, we originally used Eq. (2) with L in the denominator for the second term. The sign error was likely the reason why the Monteith method was chosen. Certainly, an update shall be considered*
10 *in the future, but it is not feasible to redo all simulations now."* The fact of the matter is that there have been some fundamental inconsistencies in the formulation of this equation, and the authors' logic of "it is not feasible to redo all simulations now" is not tenable. Ideally, the simulation would need to be done again. At the very least, the authors need to perform some test runs which demonstrate whether or not the inconsistencies in Eq. (2) make any significant difference to the results. The authors have also not clarified how the value of the zero-plane displacement height d is selected. They
15 simply state that d is constant (typically 0.7m). Looking up any atmospheric boundary-layer text book (e.g. Garratt, 1992), d is approximately 0.7 times the canopy height (or the height of the roughness element). Please check this for consistency too. We will address all matters for which we have to repeat the simulations, e.g., calculation of $R_a$ , output of dry deposition velocities, etc., at the same time. We will also check the definition of the replacement height $d$ in our formulation.

[revised manuscript text omitted]
 \frac{z-d}{z_0} - \Psi_m \frac{z_0}{L} \frac{\overline{u}(z_\text{ref}) \cdot \kappa}{\ln\left(\frac{z_\text{ref}-d_k}{z_0^k}\right) - \Psi_m\left(\frac{z_\text{ref}-d_k}{L}\right) + \Psi_m\left(\frac{z_0^k}{L}\right)}, \tag{3}$$

with the average wind speed $\overline{u}(z_\text{ref})$ at reference height, the Kármán constant $\kappa = 0.40$, the  integrated stability equation for momentum $\Psi_m$  (e.g., Garratt, 1992)

5 , a grid average Obukhov length $L$.  , deplacement height $d_k$, and roughness length $z_0^k$ ($d_k = 0.78 \cdot h_k(\text{lat})$, $z_0^k = 0.07 \cdot h_k(\text{lat})$ for forests, $d_k = 0.7 \cdot h_k(\text{lat})$, $z_0^k = 0.1 \cdot h_k(\text{lat})$ for vegetation other than forests). Taking the height of vegetation in to consideration, we have chosen the model level such that $\overline{z}_\text{ref} \approx 45$ m. Using the derived $u_*^k$ from Eq. (3), a local Obukhov length $L_k$ can be obtained from

10 (Eq. (8), Simpson et al., 2012):

$$L_k = -\frac{\rho c_p T_\text{2m} u_*^k}{\kappa g H}. \tag{4}$$

Herein, $H$ is the sensible heat flux

[revised manuscript text omitted]

$$g_{\mathrm{sto, m}}^k = g_{\mathrm{max, m}}^k \cdot f_{\mathrm{phen}}^k \cdot f_{\mathrm{light}}^k \cdot \max\left\{ f_{\min}^k, f_T^k \cdot f_D^k \cdot f_{\mathrm{SW}}^k \right\}. \tag{12}$$

The factors herein are normalized and vary within the range $0 - 1$. They account for leaf phenology ($f_{\mathrm{phen}}$), light ($f_{\mathrm{light}}$), temperature ($f_T$), water vapor pressure deficit ($f_D$), and soil water content ($f_{\mathrm{SW}}$). All factors differ with land use type $k$. For

20 clarity reasons, we drop this index in the following, as long as it is not necessary for the equation's completeness.

The temperature adjustment $f_T$  is computed from

$$f_T = \frac{T_{\mathrm{2m}} - T_{\min}}{T_{\mathrm{opt}} - T_{\min}} \cdot \left( \frac{T_{\max} - T_{\mathrm{2m}}}{T_{\max} - T_{\mathrm{opt}}} \right)^{\beta}, \tag{13}$$

with $\beta = \frac{T_{\max} - T_{\mathrm{opt}}}{T_{\mathrm{opt}} - T_{\min}}$. The parameters $T_{\min}$, $T_{\max}$ and $T_{\mathrm{opt}}$ are tabulated for various  plant functional types. All parameters  are

25 taken from Simpson et al. (2012, Tables S16, S19). Since $f_T$ turns negative outside  the range defined by $T_{\min}$, $T_{\max}$, we impose a lower limit of $0.01$ for numerical reasons.

The water vapor deficit (VPD) is proportional to the saturation partial pressure of water ($P_{\mathrm{H_2O}}^s$) and relative humidity (RH)

$$\mathrm{VPD} = P_{\mathrm{H_2O}}^s \cdot (1 - \mathrm{RH}/100). \tag{14}$$

Using tabulated values of $f_{\min}$, $D_{\min}$, $D_{\max}$, the water vapor pressure deficit penalty factor $f_D$ can be computed:

30 $$f_D = f_{\min} + (1 - f_{\min}) \cdot \frac{D_{\min} - \mathrm{VPD}}{D_{\min} - D_{\max}}. \tag{15}$$

The penalty factor with respect to available soil water (SW) $f_{SW}$ is defined as

$$f_{SW} = \begin{cases} 1 & if \quad SW \geq 0.5, \\ 2 \cdot SW & if \quad SW < 0.5. \end{cases} \tag{16}$$

SW is evaluated at a soil depths of $0.28-1$ m, which corresponds to SWVL3 in OpenIFS.

The phenology of a plant typically describes its life-cycle throughout a year, e.g., at mid latitudes and for deciduous species, it starts with the emergence of leafs in spring and ends in fall. In the  mOSaic scheme, phenology is parameterized with respect to the start of  greening season (SGS) and its end (EGS). Details about our treatment of these are given in Section 2.2.1. In summary, our adaption of the $f_{phen}$ parameterization reads as follows:

$$f_{phen} = \begin{cases} if \quad GLEN \geq 365 & 1 \quad \textit{(explicitly excluding tropics)} \\ if \quad GDAY = 0 & 0 \\ else & \begin{cases} if \quad GDAY \leq \phi_{AS} & \phi_a \\ if \quad GDAY \leq \phi_{AS} + \phi_e & \phi_b + (\phi_c - \phi_b) \cdot (GDAY - \phi_{AS})/\phi_e \\ if \quad GDAY \leq GLEN - \phi_{AE} - \phi_f & \phi_c \\ if \quad GDAY \leq GLEN - \phi_{AE} & \phi_d + (\phi_c - \phi_d) \cdot (GLEN - \phi_{AE} - GDAY)/\phi_f \\ else & \phi_d \end{cases} \end{cases} \tag{17}$$

Herein, we use the SGS and EGS derived parameters day of greening season (GDAY), the time elapsed starting at the SGS, and the total length of the greening season (GLEN), the time span between EGS and SGS. The parameters $\phi_a$, $\phi_b$, $\phi_c$, and $\phi_d$ define start or end points in the five phases of phenology in the  mOSaic scheme, while $\phi_e$, $\phi_f$, $\phi_{AS}$, and $\phi_{AE}$ control the temporal timing (Fig. 1). If GLEN is zero we are, e.g., in Arctic regions and there is no vegetation anyway, therefore $f_{phen} = 0$. Before the start of the  greening season (GDAY $= 0$) $f_{phen} = 0$. Since  this phenology is tuned to northern hemisphere (NH) mid latitudes,  it does not apply to the tropics. We therefore decided to set $f_{phen} = 1$ if GLEN is greater or equal to 365 which is the case in the tropics.

Light in the wavelength band $400-700$ nm to which the plant chlorophyll is sensitive is called photosynthetic active radiation (PAR). The integral of PAR over these wavelengths is the photosynthetic photon flux density (PPFD). The correction factor $f_{light}$ in response to varying PPFD is:

$$f_{light} = 1 - \exp(-\alpha_{light} \cdot PPFD). \tag{18}$$

[Figure]

**Figure 1.** Sketch of the five different phases in plant phenology $f_\text{phen}$ in accordance to Eq. (17).

Since $\cancel{g_\text{max}}\,\underline{g_\text{max, m}}$ in Eq. (12) is in units of $\mathrm{mmol\,s^{-1}\,m^{-2}}$, a unit conversion $\cancel{\text{is done:}}\,\underline{\text{to}\,\mathrm{m\,s^{-1}}\,\text{is necessary in our model:}}$

$$g_{\cancel{\text{sto}}}^{\cancel{\text{m}}\,k}\underline{(N)} = g_{\underline{\text{sto}}}(N)_{\text{sto, m}}^{k} \cdot R \cdot \frac{T_0}{P_0}. \tag{19}$$

Herein, $R$ is the universal gas constant.

$$\quad \cancel{G_\text{sto} = \sum_{N=0}^{N_\text{max}} f_L(N) \cdot g_\text{sto}^\text{m}(N)}$$

In the  $\underline{\text{mOSaic}}$ scheme, non-stomatal conductances are explicitly calculated for $O_3$, $SO_2$, $HNO_3$, and $NH_3$. For all other species, an interpolation between $O_3$ and $SO_2$ values is carried out.

The non-stomatal conductance for $O_3$ consists of two terms, one depending on vegetation type and one depending on the soil/surface. For $\underline{\text{each land surface types}\,k}$, we can write

$$\quad G_{\underline{\text{ns}}}^{\cancel{O_3}}(N)^{O_3,k} = \frac{\cancel{\text{SAI}(N)}\,\text{SAI}_k}{\cancel{r_\text{ext}}\,r_\text{ext}} + \frac{1}{\cancel{R_\text{inc}(N) + R_\text{gs}^{O_3}(N)}\,R_\text{inc}^k + R_\text{gs}^{O_3,k}}. \tag{20}$$

$\underline{\text{SAI}_k}$ is the surface area index for vegetation type $\underline{k}$, which is LAI plus $\underline{\text{a value that represents}}$ cuticles and others surfaces. The external leaf resistance is defined by

$$r_\text{ext} = 2000\,\mathrm{s\,m^{-1}} \cdot F_T. \tag{21}$$

Herein $F_T$ is a temperature correction factor for temperatures below $-1\,^\circ\mathrm{C}$ and $\{F_T \in \mathbb{R}\,|\,(1 \le F_T \le 2)\}$

$$\quad F_T = \underline{\exp\left(-0.2 \cdot (1 + \theta_0)\right)}\exp\left(-0.2 \cdot (1 + \theta_\text{2m})\right). \tag{22}$$

**Table 3.** Definition of growing season for crops used in the Oslo CTM3 in northern hemisphere (NH) and southern hemisphere (SH).

| | I. 1st part (days) | II. 2nd part (days) |
|---|---|---|
| NH | 90–140 | 141–270 |
| SH | 272–322 | 323–452 |

$\theta_0$ is the surface or $\theta_{2m}$ is the 2 m temperature in °C.  For most land surface types, SAI $\equiv$ LAI  Some exceptions are:

$$
\text{SAI} = \begin{cases}
\text{LAI} + 1 & if \quad forest\,/\,wetland, \\
\text{LAI} \cdot 5/3.5 & if \quad cropland, \text{1st part of growing season}, \\
\text{LAI} + 1.5 & if \quad cropland, \text{2nd part of growing season}, \\
0 & if \quad cropland, \text{winter}.
\end{cases}
\tag{23}
$$

Extending the  mOSaic scheme to the southern hemisphere,  we use the growing season for crops  defined in Table 3.

In this way, vegetation affects the conductance also by being there, not only by uptake through the stomata. The in-canopy resistance $R_{\text{inc}}$  (Erisman et al., 1994) is then modified with respect to each (vegetated) land surface type in $k$

$$
R_{\text{inc}} = b \cdot \text{SAI}(N)_k \cdot \frac{h(N,\text{lat})\; h_k(\text{lat})}{u_* \quad u_*},
\tag{24}
$$

where  $h_k(\text{lat})$ is the latitude dependent vegetation height (see explanation at the end of this section) and  $b = 14\,\text{m}^{-1}$ is an empirical constant.  The canopy resistance described in Simpson et al. (2012) does not take temperature and snow into account and is zero for non-vegetated surfaces, but we will adopt the correction previously used in the Oslo CTM3 Wesely scheme.

$$
\frac{1}{R_{\text{gs}}^{\text{O}_3}(N)} = \frac{1 - f_{\text{snow}}}{\hat{R}_{\text{gs}}^{\text{O}_3}(N)} + \frac{f_{\text{snow}}}{R_{\text{snow}}^{\text{O}_3}}.
$$

As initially mentioned, the necessary depth of snow to cover a certain type of vegetation differs. Therefore, we calculate a snow cover fraction $f_{\text{snow}}$ using the snow depth $S_D$, which is available in units of meter of water equivalent from the meteorological input data, scaled to 10 % of the vegetation height. $R_{\text{gs}}^{\text{O}_3,k}$ is tabulated. We correct for temperature by $F_T$ and for snow cover fraction:

$$
\frac{1}{R_{\text{gs}}^{\text{O}_3,k}} = \frac{1 - f_{\text{snow}}^k}{\hat{R}_{\text{gs}}^{\text{O}_3,k}} + \frac{f_{\text{snow}}^k}{R_{\text{snow}}^{\text{O}_3,k}}.
\tag{25}
$$

The

$$G_{\mathrm{ns}}^{\mathrm{O_3}} = \sum_{N=0}^{N_{\mathrm{max}}} G_{\mathrm{ns}}^{\mathrm{O_3}}(N) \cdot f_L(N).$$

bulk canopy conductance is then defined as:

$$G_c^k = \mathrm{LAI} \cdot g_{\mathrm{sto,\,m}}^k + G_{\mathrm{ns}}^k, \tag{26}$$

wherein LAI is the one-sided leaf area index taken from ISLSCP2 FASIR, $g_{\mathrm{sto}}$ the leaf-level stomatal conductance, and $G_{\mathrm{ns}}$ the bulk non-stomatal conductance.

**2.1.4 Latitude dependent vegetation height**

The vegetation height  $h_k(\mathrm{lat})$ as described by Simpson et al. (2012) is linearly decreasing with latitude between $60°$ and $74°$N. To adapt this to a global model, we  made a few additional assumptions. The tabulated height for each vegetation type  $h_k$ in the mOSaic scheme is regarded as constant at mid latitudes $(40° - 60°)$. Towards the poles, we decrease the height of each vegetation type using the same rate as described in Simpson et al. (2012). At a latitude of $74°$ a minimum height of  $3/10 \cdot h_k$ is reached and kept constant. Towards the equator, we increase the height linearly so that at a latitude of $10°$ a maximum height of  $2 \cdot h_k$ is reached which is then held constant. We also assume symmetry in both hemispheres. Presuming a typical tree height of $20\,\mathrm{m}$ at mid latitudes, this step-wise function yields a height of $8\,\mathrm{m}$ at high latitudes and $40\,\mathrm{m}$ in the tropics which is not unrealistic. For four example PFTs, results are shown in the Supplement (S..3, Fig. S4).

**2.1.5 Mapping of land surface types**

The Oslo CTM3 is configured to read land surface types from, either ISLSCP2 product from MODIS or Community Land Model (CLM) 2 categories, which have to be mapped to the  land surface types used in the  mOSaic scheme (Fig. 2). For both, MODIS and CLM 2 land surface categories, snow and ice cover is estimated from input meteorology, while  $f_L^{\mathrm{water}}$ is defined as  $1 - \sum_k f_L^k$. From the MODIS category *Barren or sparsely vegetated*, everything polward from $60°$ is defined as tundra, while everything equatorward is categorized as desert. This mapping differs from the one used in the Wesely scheme.

**2.2**

The EMEP scheme for aerosols (BC, OC, , , SOA) defines the surface deposition velocity $V_{ds}$ as

$$\frac{V_{ds}}{u_*} = \begin{cases} a_1 & \text{if} \quad L \geq 0\,\text{m}, \\ a_1 \cdot F_N \left[ 1 + \left( \frac{a_2}{25} \right)^{2/3} \right] & \text{if} \quad -25\,\text{m} < L < 0\,\text{m}, \\ a_1 \cdot F_N \left[ 1 + \left( \frac{-a_2}{L} \right)^{2/3} \right] & \text{if} \quad L \leq -25\,\text{m}, \end{cases}$$

wherein $F_N = 3$ for fine nitrate and ammonium and $F_N = 1$ for all other species. $L$ is again the Obukhov length. $a_2 = 300\,\text{m}$, while $a_1$ differs for forest and non-forest. To account for hydrophilic and hydrophobic BC/OC aerosols on dry and wet surfaces, we diverge slightly from the EMAP scheme in the definition of the parameter $a_1$. From pre-studies with an aerosol microphysic model, we find

$$a_1^{\text{hydrophob.}} = 0.025\,\text{cm}\,\text{s}^{-1} \cdot \overline{u_*}$$

and

$$a_1^{\text{hydrophil.}} = 0.2\,\text{cm}\,\text{s}^{-1} \cdot \overline{u_*},$$

[revised manuscript text omitted]

$$\mathrm{PPFD}(t_i) = \mathrm{PAR}(t_{i+1}) - \mathrm{PAR}(t_i). \tag{27}$$

For de-accumulation of the remaining time step, the best choice is subtracting the difference between $21\,\mathrm{UTC}$ and $12\,\mathrm{UTC}$ of the previous day

$$\mathrm{PPFD}(t = 00\,\mathrm{UTC}) = \mathrm{PAR}(t = 00\,\mathrm{UTC}) - [\mathrm{PAR}(t = 21\,\mathrm{UTC} - 1\,\mathrm{day}) - \mathrm{PAR}(t = 12\,\mathrm{UTC} - 1\,\mathrm{day})] \qquad (28)$$

and limit the result to positive values only. An example PAR de-accumulation for January 2nd 2005 is shown in Supplement S..5 (Figs. S5–S7). The resulting PPFD fields are still accumulated over a time period of 3 hours and should be divided by 3. A known issue in the OpenIFS (cycles $\leq$ c41r2) causes surface PAR values to be about $30\,\%$ below observations. To counter this, we decided to refrain from the division at this stage, but need to bear this in mind for later OpenIFS cycles.

**3 Evaluation**

In this section, we present results from a manifold of Oslo CTM3 model integrations testing different parameters of the  mOSaic scheme. We focus on changes in ozone  total dry deposition $\sum \mathrm{O}_3^{\mathrm{DD}}$, dry deposition velocities $v_{\mathrm{DD}}^{\mathrm{O_3}}$, concentrations in the lowermost model level $[\mathrm{O}_3](p_0)$, and tropospheric burden $\sum_{\mathrm{trop}} \mathrm{
[revised manuscript text omitted]

---

## Author Response (AR2)

**Authors' response**

To gmd-2019-21 editor decision (04 Oct 2019): Thank you for your detailed comments regarding grammar and language. We appreciate these and changed our text accordingly. In the following, we will respond to the questions in detail.

– Page 1, Abstract, line 19: Change "associated to" to "associated with" and change "resistance the ocean" to "resistance over the ocean" Changed accordingly.

– Page 2, Introduction, line 15: Replace "else" with "otherwise" Changed accordingly.

– Page 2, Introduction, line 34: Replace "observation" with "observations" Changed accordingly.

– Page 3, Introduction, line 2: Change "sides" to "sites" Changed accordingly.

– Page 3, Introduction, line 15: Worth noting that those early observations were subject to interference by other species e.g. SO2 Thank you for pointing this out. We add this remark and write: *These early observations, however, were subject to interference by other species, e.g.,* $SO_2$.

– Page 3, Introduction, line 27: Worth saying at this stage why the Luhar study gives lower estimates We add: *In particular, Luhar et al. (2018) found that the average surface resistance of ozone over ocean ($r_c = 2200\,\mathrm{s\,m^{-1}}$) is highly overestimated in most models. We add to sentence P3L14: [...] indicates much lower amounts [...] due to lower dry deposition to the oceans.*

– Page 4, Introduction, line 5: Change to "as the beginning and duration of the greening season" Changed accordingly.

– Page 4, Section 2, line 15/16: Change to "publication focussing on this is planned" Changed accordingly.

– Page 4, Section 2, line 17: Change "is given in a" to "is given at a" Changed accordingly.

– Page 7, Section 2.1.3, line 2: Remove "," after both Changed accordingly.

– Page 7, Section 2.1.3, equation 12: I'm not clear what the index m is referring to – can you explain or exclude if not necessary? The index $m$ refers to *molar*. We made this more clear. We add *molar* to the sentence preceding the equation to *[...] the leaf-level molar stomatal conductance in the mOSaic scheme [...]*. And we move the last paragraph up and change it accordingly so that it follows the equation: *The maximum molar stomatal conductance is given by $g^k_{\mathrm{max,m}}$ which is in units of $\mathrm{mmol\,s^{-1}\,m^{-2}}$. A unit conversion to $\mathrm{m\,s^{-1}}$ is necessary in our model: [...] To annotate the differing units, we use the index* m *in Eq. ([...]).*

– Page 7, Section 2.1.3, line 27: Change to "start of the greening" Changed accordingly.

– Page 9, Section 2.1.3, line 3: Change "land surface types k" to "land surface type k" Changed accordingly.

– Page 9, Section 2.1.3, line 4: Change "others" to "other" Changed accordingly.

– Page 10, Section 2.2, line 28: Change "planed" to "planned" Changed accordingly.

– Page 11, Section 2.2.1, line 10: Give web address Changed accordingly.

– Page 12, Section 2.2.1, line 5: Change "planed" to "planned" – Check for other examples Changed accordingly.

– Page 12, Section 2.2.2, line 13: Change to "spectrally" Changed accordingly.

– Page 13, Section 3, line 14: Give reference for CEDS here Changed accordingly.

– Page 13, Section 3, line 17: Can you provide global annual total? $NO_x$ emissions from soil and vegetation in the Oslo CTM3 are estimated to amount to $6.55\,\mathrm{Tg(N)\,a^{-1}}$. We add this accordingly.

- Page 14, Section 3.1, line 13: Change to "we show global distributions of the relative difference between mosaic and wesely_type for surface ozone, dry deposition velocity and total ozone dry deposition" Changed accordingly.

- Page 14, Section 3.1, line 14: Change "ozone burden" to "surface ozone" Changed accordingly.

- Page 14, Section 3.1, line 16: Change "then" to "than" Changed accordingly.

- Page 15, Section 3.2, line 5: Change to "the multi-model mean of the Task ... models" Changed accordingly.

- Page 16, Section 3.2, line 5: Change to "available" Changed accordingly.

- Page 16, Section 3.2.1, line 7: Change "in consistency with" to "consistent with" Changed accordingly.

- Page 16, Section 3.2.1, line 10: Change to "hemispheres" Changed accordingly.

- Page 16, Section 3.2.1, line 11: Change to "focussing" Changed accordingly.

- Page 16, Section 3.2.1, line 28: Change to "as the new" Changed accordingly.

- Page 16, Section 3.2.1, line 29: Change to "Arctic" Changed accordingly.

- Page 16, Section 3.2.1, line 30: Change to "strong sensitivity to" Changed accordingly.

- Page 17, Section 3.2.1, line 11: Change to "account" Changed accordingly.

- Page 17, Section 3.2.1, line 11: Change to "but do not" Changed accordingly.

- Page 17, Section 3.2.1, line 15: Change to "matches better" Changed accordingly.

- Page 17, Section 3.2.1, line 21: Change to "as a reference" – Also, in section 3.2.2 Changed accordingly.

- Page 19, Section 3.2.2, line 12: Remove "both" Changed accordingly.

- Page 19, Section 3.2.2, line 21: Change "regarding" to "for" Changed accordingly.

- Page 19, Section 3.2.2, line 23: Change "In the case of" Changed accordingly.

- Page 20, Section 3.2.2, line 19: Change to "in the Oslo" Changed accordingly.

- Page 22, Section 3.2.2, line 6: Change "responses" to "responds" Changed accordingly.

- Page 22, Section 3.3, line 16: It is unclear what you mean by "While on global average much more concise with the MACC..." We make our point more clear and change the sentences to: *Regarding the global average surface ozone concentration,* Wesely_type *(Supplement S.8) is more consistent with the MACC-reanalysis than* mOSaic*, but shows the same tendency of enhanced ozone over the continents as the latter. This enhancement is apparent mostly in the deep tropics and over the north-western Indian subcontinent which coincides with regions of high intensity in incoming UV radiation.*

- Page 23, Section 3.4, line 8: Change to "At 4 of the 6 sites" – and again, later on in the same section Changed accordingly.

- Page 25, Section 4, line 4: Correct spelling i.e. "mosaic" Changed accordingly.

- Page 25, Section 4, line 6: Change to "(with the latter" Changed accordingly.

- Page 25, Section 4, line 22: Change to "the operational" Changed accordingly.

- Page 25, Section 4, line 30: Change to "In the case of" Changed accordingly.

– Page 25, Section 4, line 30: Change "is in the order of 1 magnitude" to "is an order of magnitude" Changed accordingly.

– Page 26, Section 4, line 3: Change to "breaking" Changed accordingly.

– Page 26, Section 4, line 6: Change to "Arctic" Changed accordingly.

– Page 26, Section 4, line 11: Remove "total" Changed accordingly.

5     – Page 26, Section 4, line 14: Change to "to changes of the order of" Changed accordingly.

– Page 27: Section 4: Change all references of "arctic" to "Arctic" Changed accordingly.

– Page 27, Section 4, line 3: Change to "affect stratospheric" Changed accordingly.

– Page 27, Section 4, line 7/8: Change to "at higher" Changed accordingly.

– Author contributions: Replace reference to EMEP scheme! Changed accordingly.

[revised manuscript text omitted]
_{\mathrm{sto, m}}^k = g_{\mathrm{max, m}}^k \cdot f_{\mathrm{phen}}^k \cdot f_{\mathrm{light}}^k \cdot \max \left\{ f_{\mathrm{min}}^k, f_T^k \cdot f_D^k \cdot f_{\mathrm{SW}}^k \right\}. \tag{12}$$

The factors herein are normalized and vary within the range $0 - 1$. They account for leaf phenology ($f_{\mathrm{phen}}$), light ($f_{\mathrm{light}}$), temperature ($f_T$), water vapor pressure deficit ($f_D$), and soil water content ($f_{\mathrm{SW}}$). All factors differ with land use type $k$.

15  For clarity reasons, we drop this index in the following, as long as it is not necessary for the equation's completeness. The maximum molar stomatal conductance is given by $g_{\mathrm{max, m}}^k$ which is in units of $\mathrm{mmol \, s^{-1} \, m^{-2}}$. A unit conversion to $\mathrm{m \, s^{-1}}$ is necessary in our model:

$$g_{\mathrm{sto}}^k = g_{\mathrm{sto, m}}^k \cdot R \cdot \frac{T_0}{P_0}. \tag{13}$$

Herein, $R$ is the universal gas constant. To annotate the differing units, we use the index m in Eq. (12).

The temperature adjustment $f_T$ is computed from

$$f_T = \frac{T_{\mathrm{2m}} - T_{\mathrm{min}}}{T_{\mathrm{opt}} - T_{\mathrm{min}}} \cdot \left( \frac{T_{\mathrm{max}} - T_{\mathrm{2m}}}{T_{\mathrm{max}} - T_{\mathrm{opt}}} \right)^{\beta}, \tag{14}$$

with $\beta = \frac{T_{\mathrm{max}} - T_{\mathrm{opt}}}{T_{\mathrm{opt}} - T_{\mathrm{min}}}$. The parameters $T_{\mathrm{min}}$, $T_{\mathrm{max}}$ and $T_{\mathrm{opt}}$ are tabulated for various plant functional types. All parameters are taken from Simpson et al. (2012, Tables S16, S19). Since $f_T$ turns negative outside the range defined by $T_{\mathrm{min}}$, $T_{\mathrm{max}}$, we impose a

25  lower limit of $0.01$ for numerical reasons.

The water vapor deficit (VPD) is proportional to the saturation partial pressure of water ($P_{\mathrm{H_2O}}^s$) and relative humidity (RH)

$$\mathrm{VPD} = P_{\mathrm{H_2O}}^s \cdot (1 - \mathrm{RH}/100). \tag{15}$$

Using tabulated values of $f_{\mathrm{min}}$, $D_{\mathrm{min}}$, $D_{\mathrm{max}}$, the water vapor pressure deficit penalty factor $f_D$ can be computed:

$$f_D = f_{\mathrm{min}} + (1 - f_{\mathrm{min}}) \cdot \frac{D_{\mathrm{min}} - \mathrm{VPD}}{D_{\mathrm{min}} - D_{\mathrm{max}}}. \tag{16}$$

[Figure]

**Figure 1.** Sketch of the five different phases in plant phenology $f_{\text{phen}}$ in accordance to Eq. (18).

The penalty factor with respect to available soil water (SW) $f_{\text{SW}}$ is defined as

$$f_{\text{SW}} = \begin{cases} 1 & \text{if} \quad \text{SW} \geq 0.5, \\ 2 \cdot \text{SW} & \text{if} \quad \text{SW} < 0.5. \end{cases} \tag{17}$$

SW is evaluated at a soil depths of $0.28 - 1\,\text{m}$, which corresponds to SWVL3 in OpenIFS.

The phenology of a plant typically describes its life-cycle throughout a year, e.g., at mid latitudes and for deciduous species,
5  it starts with the emergence of leafs in spring and ends in fall. In the mOSaic scheme, phenology is parameterized with respect
to the start of the greening season (SGS) and its end (EGS). Details about our treatment of these are given in Section 2.2.1. In
summary, our adaption of the $f_{\text{phen}}$ parameterization reads as follows:

$$f_{\text{phen}} = \begin{cases} \text{if} \quad \text{GLEN} \geq 365 & 1 \quad \textit{(explicitly excluding tropics)} \\ \text{if} \quad \text{GDAY} = 0 & 0 \\ \text{else} & \begin{cases} \text{if} \quad \text{GDAY} \leq \phi_{\text{AS}} & \phi_a \\ \text{if} \quad \text{GDAY} \leq \phi_{\text{AS}} + \phi_e & \phi_b + (\phi_c - \phi_b) \cdot (\text{GDAY} - \phi_{\text{AS}})/\phi_e \\ \text{if} \quad \text{GDAY} \leq \text{GLEN} - \phi_{\text{AE}} - \phi_f & \phi_c \\ \text{if} \quad \text{GDAY} \leq \text{GLEN} - \phi_{\text{AE}} & \phi_d + (\phi_c - \phi_d) \cdot (\text{GLEN} - \phi_{\text{AE}} - \text{GDAY})/\phi_f \\ \text{else} & \phi_d \end{cases} \end{cases} \tag{18}$$

[revised manuscript text omitted]